# Assembly of neuron- and radial glial-cell-derived extracellular matrix molecules promotes radial migration of developing cortical neurons

Ayumu Mubuchi[1], Mina Takechi[2], Shunsuke Nishio[3], Tsukasa Matsuda[3], Yoshifumi Itoh[4], Chihiro Sato[2,5,6], Ken Kitajima[2,5,6], Hiroshi Kitagawa[7], Shinji Miyata[1]*

[1]Graduate School of Agriculture, Tokyo University of Agriculture and Technology, Fuchu, Japan; [2]Graduate School of Bioagricultural Sciences, Nagoya University, Nagoya, Japan; [3]Faculty of Food and Agricultural Sciences, Fukushima University, Fukushima, Japan; [4]Kennedy Institute of Rheumatology, University of Oxford, Oxford, United Kingdom; [5]Bioscience and Biotechnology Center, Nagoya University, Nagoya, Japan; [6]Institute for Glyco-core Research, Nagoya University, Nagoya, Japan; [7]Laboratory of Biochemistry, Kobe Pharmaceutical University, Kobe, Japan

*For correspondence:
smiyata@go.tuat.ac.jp

Competing interest: The authors declare that no competing interests exist.

**Abstract** Radial neuronal migration is a key neurodevelopmental event for proper cortical laminar organization. The multipolar-to-bipolar transition, a critical step in establishing neuronal polarity during radial migration, occurs in the subplate/intermediate zone (SP/IZ), a distinct region of the embryonic cerebral cortex. It has been known that the extracellular matrix (ECM) molecules are enriched in the SP/IZ. However, the molecular constitution and functions of the ECM formed in this region remain poorly understood. Here, we identified neurocan (NCAN) as a major chondroitin sulfate proteoglycan in the mouse SP/IZ. NCAN binds to both radial glial-cell-derived tenascin-C (TNC) and hyaluronan (HA), a large linear polysaccharide, forming a ternary complex of NCAN, TNC, and HA in the SP/IZ. Developing cortical neurons make contact with the ternary complex during migration. The enzymatic or genetic disruption of the ternary complex impairs radial migration by suppressing the multipolar-to-bipolar transition. Furthermore, both TNC and NCAN promoted the morphological maturation of cortical neurons in vitro. The present results provide evidence for the cooperative role of neuron- and radial glial-cell-derived ECM molecules in cortical development.

## eLife assessment

The **solid** study addresses the role of extracellular matrix (ECM) in neuronal migration. The authors showed that the interaction between the ternary complex formed by tenascin-C, the chondroitin sulfate proteoglycan neurocan, and hyaluronic acid is **important** for the multipolar to bipolar transition in the intermediate zone (IZ) of the developing cortex

## Introduction

The six-layered laminar structure of the mammalian cerebral neocortex is achieved via the coordinated radial migration of neurons from their place of birth in the ventricular zone (VZ) to their final destination in the cortical plate (CP; *Barnes and Polleux, 2009*; *Taverna et al., 2014*). Radial glial cells residing in the VZ are embryonic neural stem cells and a major source of neurons (*Kriegstein*

and Alvarez-Buylla, 2009). During the neurogenesis period (from embryonic day (E) 12 to E18 in mice), radial glial cells divide asymmetrically to give rise to one self-renewing radial glial cell and one intermediate progenitor cell. Intermediate progenitor cells detach from the apical surface, lose their polarity, and further divide symmetrically to produce a pair of neurons above the VZ (Noctor et al., 2004). Newly born immature neurons transiently exhibit multipolar shapes with multiple neurites and move in random directions (Noctor et al., 2004; Tabata and Nakajima, 2003). Neurons then change their shape from multipolar to bipolar, extending leading and trailing processes in the subplate/intermediate zone (SP/IZ) (Hatanaka and Yamauchi, 2013; Jossin and Cooper, 2011; Namba et al., 2014; Ohtaka-Maruyama et al., 2018). Bipolar neurons attach to the radial fibers elongated from radial glial cells and migrate radially along these fibers toward their final destination in the CP. The leading and trailing processes of bipolar neurons eventually become the apical dendrite and axon. Therefore, radial glial cells play important dual roles in both producing neurons and guiding them to their proper locations.

The multipolar-to-bipolar transition is a critical step in establishing neuronal polarity. The intrinsic mechanisms underlying neuronal polarity have been investigated for a long time using primary cultured neurons in vitro, and the importance of cytoskeletal molecules, signaling cascades, and transcription factors has been demonstrated (Jossin, 2020; Namba et al., 2015). Polarity formation in vivo may differ from that in vitro because it occurs in a complex three-dimensional environment generated by the extracellular matrix (ECM), secreted factors, and neighboring cells. The ECM provides structural support to guarantee tissue integrity and regulates neuronal differentiation, maturation, and synaptic function in the brain (Barros et al., 2011; Faissner and Reinhard, 2015; Franco and Müller, 2011). The brain ECM is characterized by a predominance of chondroitin sulfate proteoglycans (CSPGs; Fawcett et al., 2022; Maeda et al., 2011; Miyata and Kitagawa, 2017; Reinhard et al., 2016; Zimmermann and Dours-Zimmermann, 2008). It has long been known that CSPGs are enriched in the SP/IZ (Fukuda et al., 1997; Hoerder-Suabedissen and Molnár, 2015; Miller et al., 1995). Notably, the multipolar-to-bipolar transition occurs in the SP/IZ, a cell-sparse area rich in CSPGs. However, the molecular constitution and functions of the ECM formed in this region remain poorly understood.

CSPGs expressed in the brain, including neurocan (NCAN), aggrecan (ACAN), versican, and brevican, exhibit shared structural domains (Yamaguchi, 2000). The N-terminal domain interacts with hyaluronan (HA), a large linear polysaccharide chain that reaches several million Da in size (Day and Prestwich, 2002). The central region shows an extended structure and contains covalently bound chondroitin sulfate chains (Zimmermann and Dours-Zimmermann, 2008). Previous studies have demonstrated the critical roles of chondroitin sulfate chains in neural stem/progenitor cell proliferation and neuronal migration (Ishii and Maeda, 2008; Sirko et al., 2007; von Holst et al., 2006). The C-terminal domain of CSPGs binds to tenascins, a group of multimeric glycoproteins (Aspberg et al., 1997; Yamaguchi, 2000). Many studies have reported that tenascin-C (TNC) expressed by radial glial cells in the embryonic brain plays a crucial role in controlling cell proliferation and differentiation in the VZ (see reviews by Faissner et al., 2017; Faissner and Reinhard, 2015). While the involvement of TNC in the stem cell niche is well-established, its role in neuronal migration remains relatively unexplored. For instance, a previous study has demonstrated that anti-TNC antibodies inhibit the migration of cerebellar granule cells in explant cultures (Husmann et al., 1992).

We previously reported that HA accumulated in the SP/IZ of the developing mouse cerebral cortex (Takechi et al., 2020). Pharmacological inhibition of HA synthesis suppressed neurite outgrowth in cultured neurons in vitro. Nevertheless, the interplay between HA, CSPGs, and tenascins within a common pathway to regulate embryonic cortical development remains uncertain. In the present study, we aimed to investigate whether CSPGs, HA, and tenascins form a complex in the developing cerebral cortex and, if so, to determine the potential significance of this complex in neuronal migration and the multipolar-to-bipolar transition in vivo. We demonstrated that NCAN, a major CSPG produced by developing cortical neurons, is bound to HA in the SP/IZ in vivo. NCAN also interacted with radial glial-cell-derived TNC, forming a ternary complex of HA, NCAN, and TNC. Developing cortical neurons made contact with the ternary complex during migration. The enzymatic or genetic disruption of the ternary complex retarded radial migration by suppressing the multipolar-to-bipolar transition. The present results provide evidence for the cooperative role of neuron- and radial glial-cell-derived ECM molecules in cortical development.

## Results

### NCAN is a major CSPG produced by developing cortical neurons

We attempted to identify the core proteins of CSPGs in the developing mouse brain using an antibody recognizing the neoepitope generated at the non-reducing terminus of chondroitin sulfate chains after digestion with chondroitinase ABC (*Figure 1a*; *Caterson, 2012*). Immunoblot analysis of the mouse cerebral cortex at E18.5 indicated that a band at 130 kDa represented a major CSPG detected by the anti-neo epitope antibody (*Figure 1b*). However, the developing mouse cerebral cortex contained other minor CSPGs. The lack of staining from the undigested cerebral cortex confirmed the specificity of the antibody. In-gel tryptic digestion followed by a MALDI-TOF/TOF analysis identified the 130 kDa CSPG as an N-terminal portion NCAN (*Figure 1c*). Full-length >250 kDa NCAN is reportedly cleaved to generate the N-terminal 130 kDa fragment and C-terminal 150 kDa fragment (*Figure 1d*; *Rauch et al., 1992*). Using an antibody to the N-terminal portion of NCAN, we found that the full-length and the N-terminal fragment of NCAN were both detected in the developing cerebral cortex (*Figure 1e and f*). In contrast, only the cleaved fragment was detected in the adult cerebral cortex, indicating the developmental stage-dependent proteolytic processing of NCAN (*Figure 1e and f*). Quantitative RT-PCR analysis showed high expression levels of *Ncan* during the late embryonic and early postnatal stages (*Figure 1g*).

We examined whether developing cortical neurons produce NCAN by immunoblot analysis of primary cultured cortical neurons. The full-length and the N-terminal fragment of NCAN were gradually secreted into the culture medium (*Figure 1h*). This result suggested the secretion of NCAN by developing neurons; however, we cannot rule out the involvement of coexisting glial cells in the culture system. To investigate the expression of *Ncan* mRNA during radial migration in vivo, we labeled radial glial cells in the VZ with GFP through in utero electroporation at E14.5 (*Figure 1i*, *Figure 1—figure supplement 1*). Quantitative RT-PCR analysis of GFP-positive cells isolated 0.6, 1, 2, and 4 days after labeling demonstrated that *Ncan* mRNA expression increased 2 days after labeling and was thereafter maintained (*Figure 1j*). These results indicated that developing cortical neurons themselves express *Ncan* mRNA and are capable of secreting NCAN.

### NCAN forms a pericellular matrix with HA around developing neurons

We investigated the distribution of *Ncan* mRNA in the developing cerebral cortex by in situ hybridization analysis. The widespread distribution of *Ncan* mRNA throughout the cerebral cortex indicated that NCAN production involves developing neurons as well as other cell populations, including radial glial cells (*Figure 2a*). In contrast to *Ncan* mRNA, NCAN protein densely accumulated in cell-sparse zones, such as the SP/IZ and marginal zone (MZ), but was less abundant in cell-dense regions, such as the CP and VZ (*Figure 2b*). This discrepancy in the *Ncan* mRNA and NCAN protein distribution pattern suggested a mechanism through which secreted NCAN protein localizes to the SP/IZ and MZ.

We noticed that the localization pattern of NCAN was similar to that of HA, which we previously reported (*Figure 2c*; *Takechi et al., 2020*). Furthermore, NCAN and HA co-localized in cortical neuronal cultures as punctate signals at adhesion sites between the neuron and culture substrate, indicating that NCAN formed a pericellular matrix with HA (*Figure 2d*). To directly examine the binding between NCAN and HA, we pulled down endogenous HA and its interacting proteins using a biotinylated HA-binding protein (b-HABP), followed by immunoblotting with the antibody to the N-terminal portion of NCAN (*Figure 2e*). This analysis confirmed the interaction between NCAN and endogenous HA (*Figure 2f*). NCAN did not precipitate without b-HABP. The precipitation of NCAN depended on intact HA because NCAN disappeared from precipitates by digestion with hyaluronidase from *Streptomyces hyalurolyticus*, which specifically degrades HA and does not act on other glycosaminoglycans (*Ohya and Kaneko, 1970*). In addition, exogenously added biotinylated HA (b-HA) is also bound to the N-terminal portion of NCAN (*Figure 2g*). To verify the association between NCAN and HA by reciprocal co-immunoprecipitation, we recombinantly expressed the full-length, N- and C-terminal halves of NCAN fused with GFP (*Figure 2—figure supplement 1a*). After incubating with HA, we pulled down GFP-fused NCAN with anti-GFP nanobody-conjugated resin. Quantification of HA by ELISA demonstrated that the full-length and N-terminal half of NCAN precipitated a larger amount of HA than the C-terminal half (*Figure 2—figure supplement 1b*), supporting the binding between the N-terminal portion of NCAN and HA.

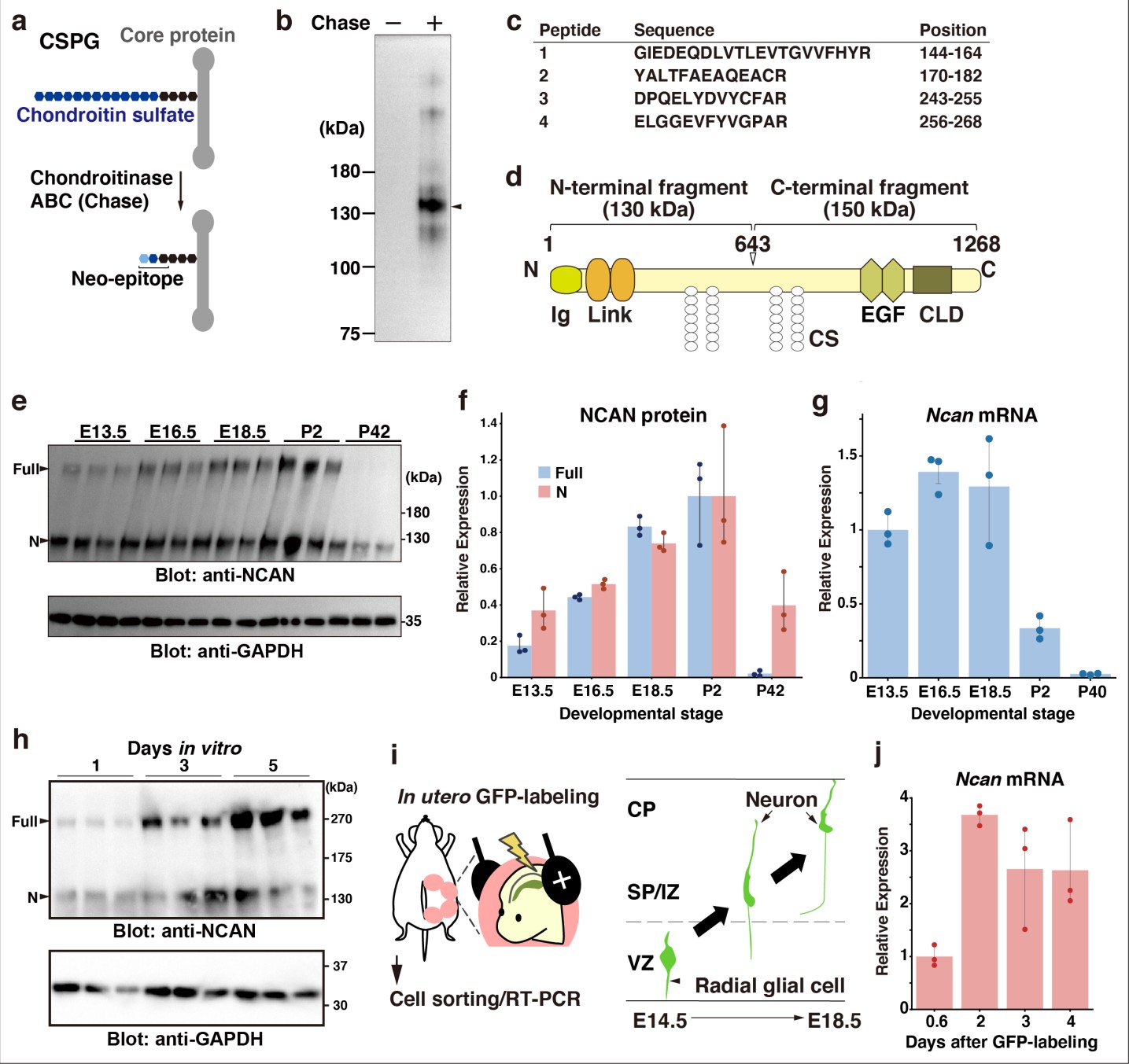

**Figure 1.** NCAN is a major CSPG produced by developing cortical neurons. (**a**) Detection of the neoepitope of CSPGs after digestion with chondroitinase ABC (Chase). (**b**) Immunoblotting of the neoepitope from undigested (Chase -) and digested (Chase +) cerebral cortex lysates prepared at E18.5. The arrowhead indicates the major CSPG at 130 kDa. (**c**) List of peptide fragments identified from the 130 kDa band. Positions indicate the amino acid number in full-length NCAN. (**d**) Domain structure of mouse NCAN. Ig: immunoglobulin-like domain, Link: hyaluronan-binding link module, EGF: epidermal growth factor-like repeat, CLD: C-type lectin domain, CS: chondroitin sulfate chain. (**e, f**) NCAN expression in the developing cerebral cortex from E13.5 to postnatal day (P) 42. The arrowheads indicate the full-length and N-terminal fragment of NCAN. Values in (**f**) are normalized to GAPDH and represented relative to P2. N=3 for each point. Mean ± SD. (**g**) Quantitative RT-PCR analysis of *Ncan* mRNA in the developing cerebral cortex. Values are normalized to *Gapdh* and represented relative to E13.5. N=3 for each point. Mean ± SD. (**h**) Immunoblot analysis of NCAN in the cultured medium of primary cultured cortical neurons 1, 3, and 5 days after plating. (**i**) Experimental model for in utero labeling. Radial glial cells in the VZ were labeled with GFP on E14.5. GFP-positive cells were isolated 0.6–4 days later. (**j**) Expression of *Ncan* mRNA in GFP-labeled cells isolated on the indicated days after in utero electroporation. Values are normalized to *Gapdh* and represented relative to 0.6 days. N=3 for each point. Mean ± SD.

The online version of this article includes the following source data and figure supplement(s) for figure 1:

*Figure 1 continued on next page*

*Figure 1 continued*

**Source data 1.** Numerical source data and original blots for *Figure 1*.

**Figure supplement 1.** Location of GFP-positive cells 0.6, 2, and 4days after in utero electroporation.

## Radial glial-cell-derived TNC interacts with the C-terminal portion of NCAN

To further screen the interacting partners of NCAN, we pulled down proteins in the embryonic mouse brain lysate with GFP-fused full-length NCAN immobilized to resin (*Figure 3a*). LC-MS/MS analysis following trypsin digestion identified several proteins, including NCAN itself (*Table 1*). Among the identified proteins, only TNC exhibited specific binding to NCAN. In contrast, intracellular proteins such as tubulin and actin were found to be non-specific proteins detected even from the negative control resin. Co-immunoprecipitation followed by immunoblot analysis showed that the C-terminal half, but not the N-terminal half of NCAN, is bound to TNC (*Figure 3b*). Interactions between NCAN and TNC were dependent on a divalent cation but not on chondroitin sulfate chains on NCAN consistent with previous findings (*Aspberg et al., 1997*; *Rauch et al., 1997*). The expression of TNC at mRNA and protein levels peaked during late embryonic brain development, similar to the expression pattern of NCAN (*Figure 3c–e*). In situ hybridization analysis indicated that in the developing cerebral cortex at E16.5, *Tnc* mRNA was confined to the VZ where the cell bodies of radial glia cells reside (*Figure 3f*), consistent with previous findings (*Faissner et al., 2017*; *Garcion et al., 2004*). While some astrocytes express TNC during gliogenesis (E18 or later), it has been documented that radial glial cells are the primary source of TNC in earlier stages (*Faissner et al., 2017*). However, TNC protein was not restricted to the VZ but was also observed in the SP/IZ and MZ, similar to the localization of NCAN (*Figure 3g*).

To predict the structure of the NCAN-TNC complex, we performed AlphaFold2 multimer analysis (*Figure 4a and b*, *Figure 4—figure supplement 1a*). In this modeled complex, a C-type lectin domain (CLD) of NCAN interacts with the 4th and 5th fibronectin type-III domains (FNIII) of TNC, which is consistent with a previously reported structure of aggrecan (ACAN) and tenascin-R (TNR; *Figure 4—figure supplement 1b*; *Lundell et al., 2004*). Amino acid residues in the interface of NCAN-TNC and ACAN-TNR complex were highly conserved (*Figure 4c and d*). An Asn residue (Asn1148 in mNCAN and Asn2022 in rACAN), which interacts with $Ca^{2+}$ ions, was also conserved. Sequence alignments of fourteen FNIII of TNC indicated that the amino acid residues in FNIII-4 involved in the binding to NCAN are not well conserved among other FNIII (*Figure 4—figure supplement 1c*). From these results, we proposed that NCAN, HA, and TNC cooperatively assemble a supramolecular complex as follows: (1) NCAN derived from developing neurons and radial glial cells binds to multiple sites on a long chain of HA via the N-terminal Link modules; (2) The hexamer of TNC secreted from radial glial cells cross-links NCAN through the interaction between the FNIII-4 domain of TNC and the CLD of NCAN (*Figure 4e*).

## HA, NCAN, and TNC form the ternary complex in the developing cerebral cortex

To investigate the formation of a ternary complex of HA, NCAN, and TNC in vivo, we analyzed the localization of these molecules in the embryonic mouse cerebral cortex. HA, NCAN, and TNC were co-localized in the upper part of the SP/IZ and MZ of the developing cerebral cortex at E17.5 (*Figure 5a*). High-magnification views showed the aggregate-like deposition of these molecules in the SP/IZ (*Figure 5b*). Co-localization of these three molecules was rarely observed in the CP or VZ. To investigate whether the ternary complex interacts with the neuronal surface in vivo, we labeled developing cortical neurons by transfecting a plasmid expressing a membrane-bound form of GFP at E14.5. Three days after in utero electroporation, GFP-positive neurons exhibited a bipolar morphology in the upper SP/IZ and CP (*Figure 5c and d*), as previously reported (*Tabata and Nakajima, 2003*). Z-stack projections of confocal images revealed that the ternary complex frequently contacted the surface of migrating neurons (*Figure 5c and d*). Orthogonal views of 3D images showed the extracellular deposition of the ternary complex surrounding the cell body and neurites of bipolar neurons (*Figure 5c and d*).

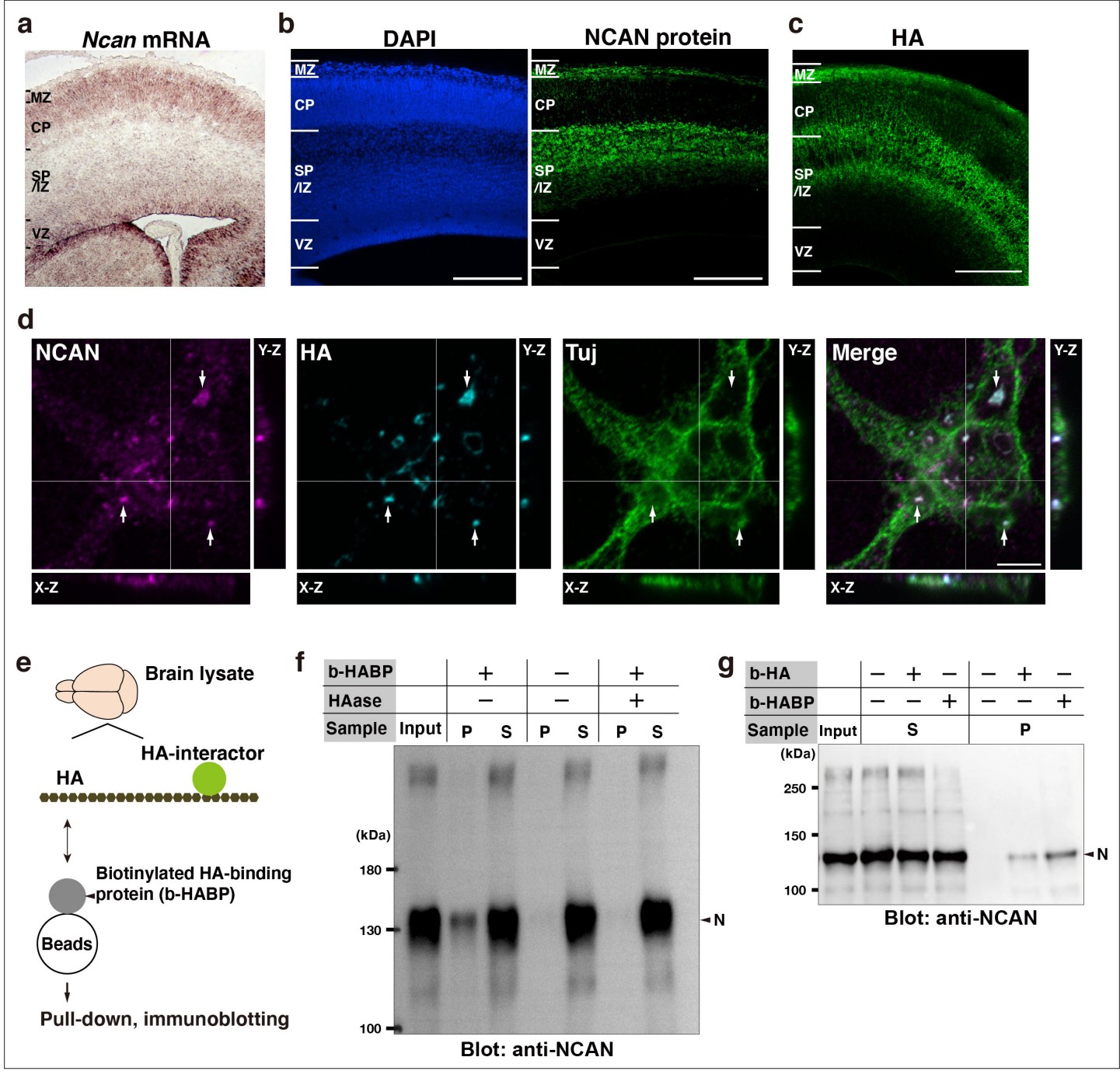

**Figure 2.** Neuron-derived NCAN forms a pericellular matrix with HA. (**a**) In situ hybridization analysis of *Ncan* mRNA on the E16.5 cerebral cortex. (**b**) Localization of NCAN protein (green) in the E16.5 cerebral cortex. Nuclei were counterstained with DAPI (blue). (**c**) Distribution of HA (green) visualized by the biotinylated HA-binding protein (b-HABP) in the E16.5 cerebral cortex. (**d**) High-magnification views of a Tuj-1-positive primary cultured cortical neuron (green) 5 days after plating. White arrows indicate the co-localization of NCAN (magenta) and HA (cyan). Orthogonal projections in the X-Z and Y-Z planes taken along the white lines showed the localization of NCAN and HA at the adhesion sites between the neuron and culture substrate. (**e**) Schema of the pull-down assay for analyzing binding between endogenous HA and its interactors. (**f**) Immunoblotting of the input, precipitate (P), and supernatant (S) with an anti-NCAN antibody. NCAN was precipitated with HA by adding b-HABP (b-HABP +). NCAN was not precipitated without b-HABP (b-HABP -) or after digestion with hyaluronidase (HAase +). (**g**) Co-precipitation of NCAN with exogenously added biotinylated HA (b-HA +) or endogenous HA (b-HABP +). Scale bars represent 200 µm (**a–c**) and 5 µm (**d**).

The online version of this article includes the following source data and figure supplement(s) for figure 2:

**Source data 1.** Original blots for *Figure 2*.

*Figure 2 continued on next page*

*Figure 2 continued*

**Figure supplement 1.** Pull-down assay of HA with recombinant GFP-fused NCAN.

**Figure supplement 1—source data 1.** Numerical source data and original blots for *Figure 2—figure supplement 1*.

## Defects in NCAN and TNC retards neuronal migration

To elucidate the role of the ternary complex during cortical development, we generated double knockout (DKO) mice for NCAN and TNC. Neither TNC nor NCAN was detected in the DKO mice, and the brain weights were not different compared to wild-type (WT) mice (*Figure 6—figure supplement 1*). Since the migratory patterns of neurons differ between the lateral and medial cortices, we took advantage of Flash Tag technology, in which an intraventricular injection of fluorescent dyes allows for the pulse labeling of progenitors in contact with the ventricle across the brain (*Govindan et al., 2018*; *Telley et al., 2016*; *Yoshinaga et al., 2021*). Radial glial cells in the M-phase were labeled at E14.5, and their progeny was analyzed 2 days later at E16.5. In the lateral cortex of WT mice, many labeled cells were migrating in the upper and lower SP/IZ, and a smaller population had

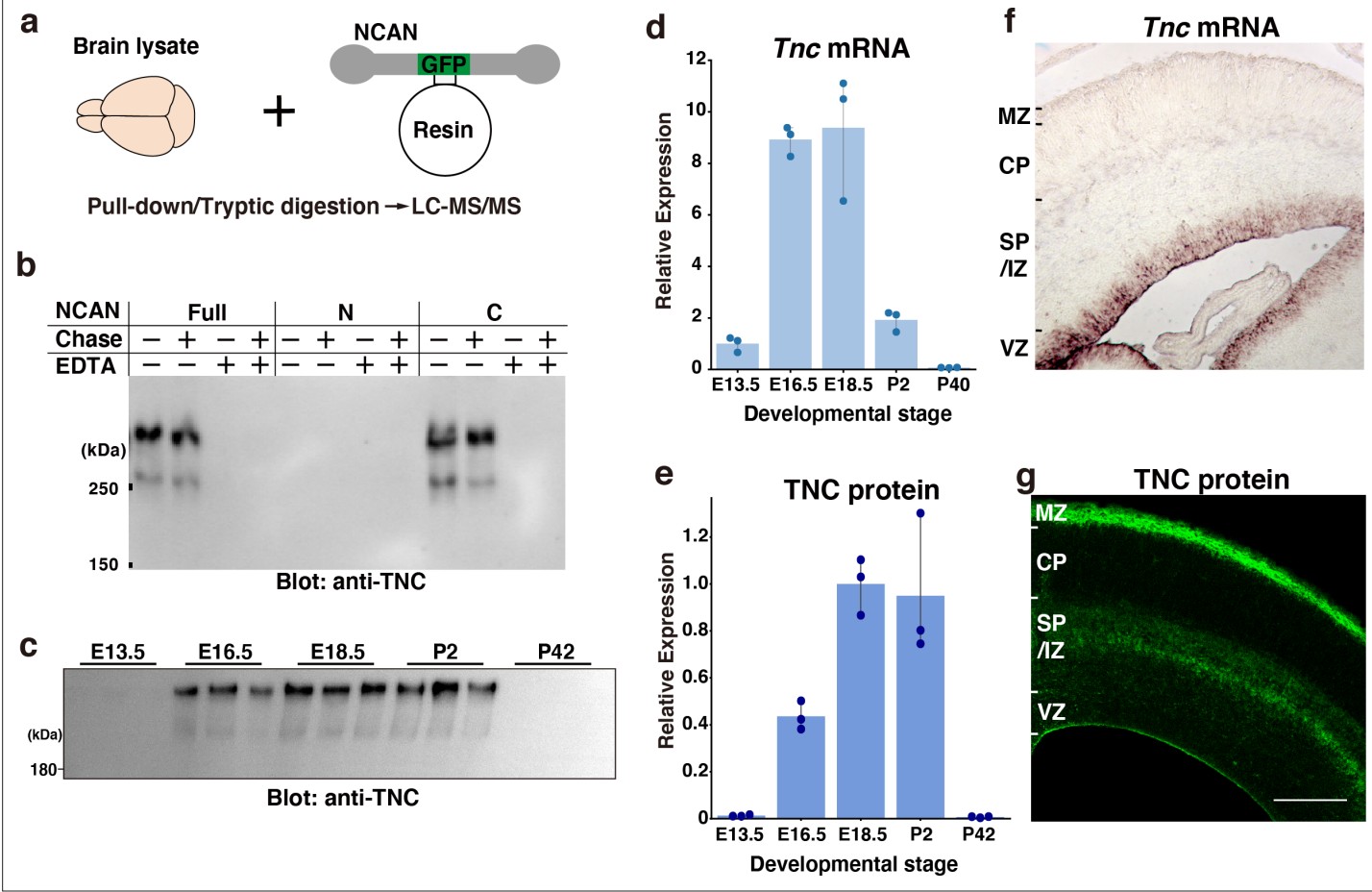

**Figure 3.** Screening of the interacting partners of NCAN. (**a**) Schematic of the pull-down assay to identify NCAN-interacting partners. (**b**) Co-precipitation of TNC with the full-length and C-terminal half of NCAN. Interactions between NCAN and TNC disappeared following the addition of EDTA but were not affected by Chase digestion. The N-terminal half of NCAN did not bind to TNC. (**c, e**) TNC expression in the developing cerebral cortex from E13.5 to postnatal day (P) 42. Values in (**e**) are normalized to GAPDH and represented relative to E18.5. The same brain lysate as in *Figure 1e* were analyzed. The values for GAPDH were calculated based on the data presented in *Figure 1e*. N=3 for each point. Mean ± SD. (**d**) Quantitative RT-PCR analysis of *Tnc* mRNA in the developing cerebral cortex. Values are normalized to *Gapdh* and represented relative to E13.5. N=3 for each point. Mean ± SD. (**f, g**) In situ hybridization analysis of *Tnc* mRNA (**f**) and immunohistochemical localization of TNC protein (**g**) in the E16.5 cerebral cortex. Scale bar represents 200 μm.

The online version of this article includes the following source data for figure 3:

**Source data 1.** Numerical source data and original blots for *Figure 3*.

**Table 1.** List of proteins identified from GFP-NCAN and negative control resins.

| GFP-NCAN resin | | Negative control resin | |
| --- | --- | --- | --- |
| Protein name | No. of peptide | Protein name | No. of peptide |
| Neurocan core protein | 34 | Tubulin beta-2B chain | 21 |
| Tenascin | 22 | Tubulin beta-5 chain | 17 |
| Tubulin alpha-1A chain | 18 | Tubulin beta-3 chain | 15 |
| Tubulin beta-2B chain | 17 | Tubulin beta-6 chain | 15 |
| Tubulin beta-5 chain | 16 | Tubulin alpha-1A chain | 13 |
| Actin, cytoplasmic 2 | 11 | Actin, cytoplasmic 2 | 10 |
| Actin, cytoplasmic 1 | 9 | Elongation factor 1-alpha 1 | 5 |
| Tubulin beta-6 chain | 8 | Macrophage migration inhibitory factor | 5 |
| Elongation factor 1-alpha 1 | 6 | Crk-like protein | 4 |
| Crk-like protein | 4 | Keratin, type II cytoskeletal 1 | 4 |
| Profilin-2 | 4 | Peroxiredoxin-1 | 3 |
| Eukaryotic translation initiation factor 3 subunit L | 4 | L-lactate dehydrogenase A chain | 3 |
| Histone deacetylase 6 | 4 | Keratin, type I cytoskeletal 10 | 3 |
| 40 S ribosomal protein SA | 3 | Hemoglobin subunit alpha | 3 |
| Macrophage migration inhibitory factor | 3 | Hemoglobin subunit beta-1 | 3 |
| Eukaryotic translation initiation factor 3 subunit E | 3 | Keratin, type II cytoskeletal 1b | 3 |
| Keratin, type II cytoskeletal 1b | 3 | | |

already reached the CP (*Figure 6a and b*). In DKO mice, most cells resided beneath the lower SP/IZ, and very few reached the upper SP/IZ and CP. Bin analysis showed that DKO mice had a significantly lower percentage of labeled cells in the upper layers (Bin 1 and 2) and more cells in the lower layers (Bin 4 and 5) than WT mice (*Figure 6b*). In the medial cortex of WT mice, neurons reached the CP more quickly than in the lateral cortex (*Figure 6c*; *Yoshinaga et al., 2021*). DKO mice exhibited slower neuronal migration in the medial cortex than WT mice, suggesting that the ternary complex facilitated the entry of neurons into the CP in a wide range of cortical regions. However, delayed migration in DKO mice was mostly restored after 3 days of labeling (*Figure 6—figure supplement 2a, b*).

To investigate the laminar organization of the postnatal cerebral cortex, we analyzed the distribution of NeuN-positive postmitotic neurons in DKO mice at 2 weeks of age. No notable abnormalities were observed in the laminar structure of DKO mice (*Figure 6—figure supplement 3a, b*). Additionally, the laminar distribution of Ctip2-positive deep layer neurons showed no significant differences between WT and DKO mice (*Figure 6—figure supplement 3a, c*). These results suggest that while the ternary complex is necessary for proper neuronal migration, some compensatory mechanisms may mitigate its effects in the postnatal brain. To assess whether the loss of NCAN and TNC affects stem cells/progenitors in the VZ, we compared the number of Pax6-positive radial glial cells and Tbr2-positive intermediate progenitor cells. Immunohistochemical analysis showed no significant differences between WT and DKO mice (*Figure 6—figure supplement 4a*). Additionally, the morphology of nestin-positive radial fibers was indistinguishable between WT and DKO mice (*Figure 6—figure supplement 4b, c*). These results suggested that the delay in neuronal migration was not due to the decreased proliferation of radial glial cells in DKO mice.

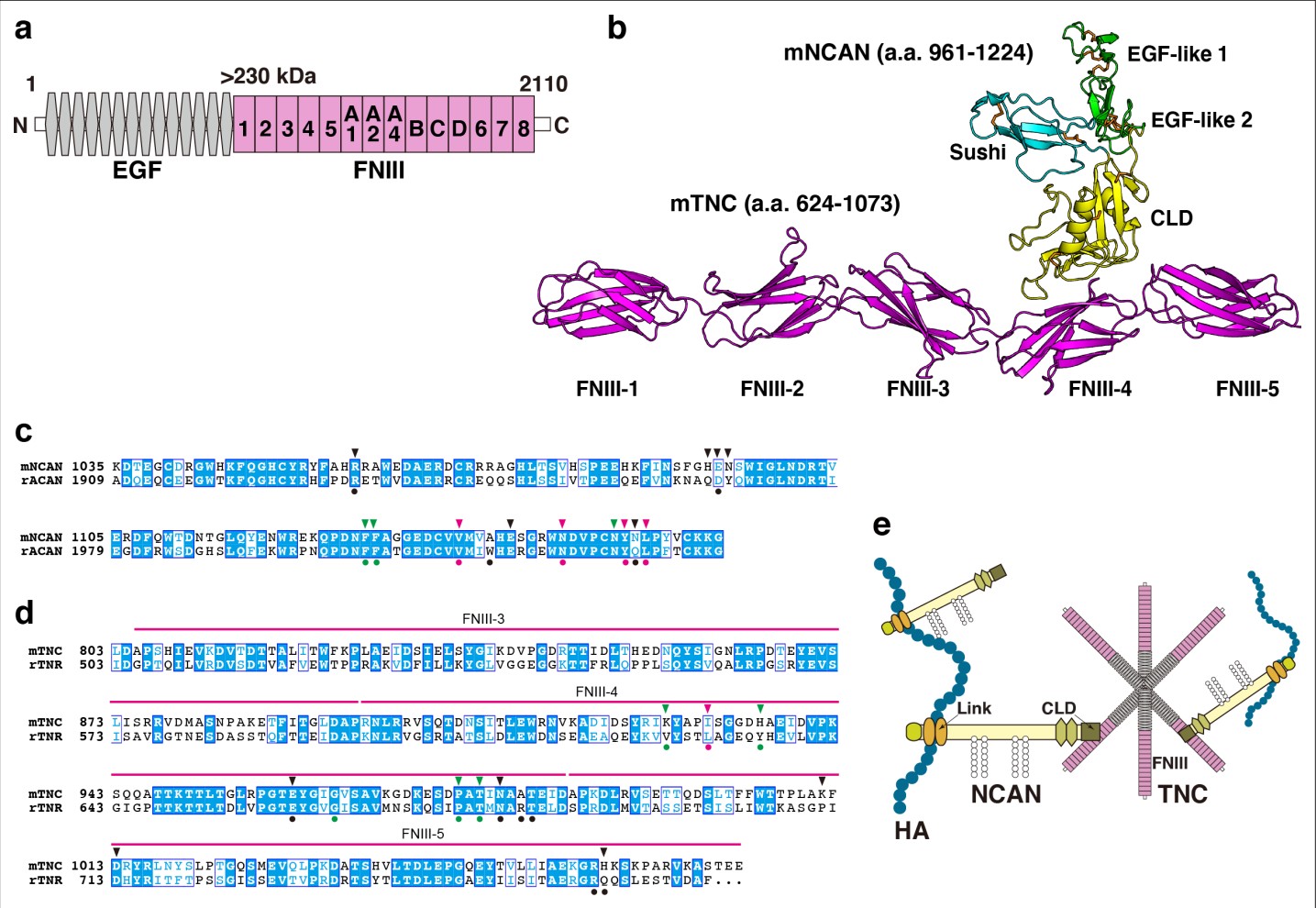

**Figure 4.** Alphafold2 prediction of the NCAN-TNC complex. (**a**) Domain structure of mouse (m) TNC. EGF: epidermal growth factor-like repeat, FNIII: fibronectin type-III domain. (**b**) Five predicted complex models of the mNCAN and mTNC were generated using AlphaFold2 multimer implemented in ColabFold, and the best-predicted complex was shown. (**c, d**). Sequence alignments of CLD of mNCAN and rat (**r**) ACAN (**c**) and FNIII-3–5 of mTNC and rTNR (**d**). Circles under alignments indicate the key residues of the rACAN-rTNR complex. Triangles over alignments indicate the residues in the interface of mNCAN-mTNC. Green and magenta circles/triangles are the residues involved in the interaction between L4 loop of CLD and βC, F, G strands of FNIII-4 and β6, 7 strands of CLD and CC′ loop of FNIII-4, respectively. Black circles/triangles indicate residues involved in the interaction between CLD and FNIII-4–5 linker region/FG loop of FNIII-5. (**e**) Model for forming the ternary complex of NCAN, HA, and TNC.

The online version of this article includes the following figure supplement(s) for figure 4:

**Figure supplement 1.** Alphafold2 prediction of NCAN-TNC complex.

## Single deletion of NCAN or TNC results in mild abnormalities in neuronal migration

To investigate whether the loss of either NCAN or TNC alone would result in delayed neuronal migration similar to that observed in the DKO mice, we analyzed single KO mice lacking NCAN or TNC. In NCAN KO mice, a significantly lower percentage of labeled cells resided in the upper layer (Bin2), and more cells remained in the lower layer (Bin5) than in WT mice (*Figure 7a*). In contrast, the impact of a single deletion of TNC on neuronal cell migration was very limited. Although TNC KO mice exhibited a tendency to have a higher proportion of labeled cells in the lower layer (Bin4) than in WT mice, this did not reach statistical significance (*Figure 7a*). The delay in neuronal migration observed in the single KO mice was milder when compared to that observed in DKO mice (*Figure 6a–c*), suggesting that simultaneous deletion of both NCAN and TNC is necessary for a more pronounced impairment in neuronal cell migration.

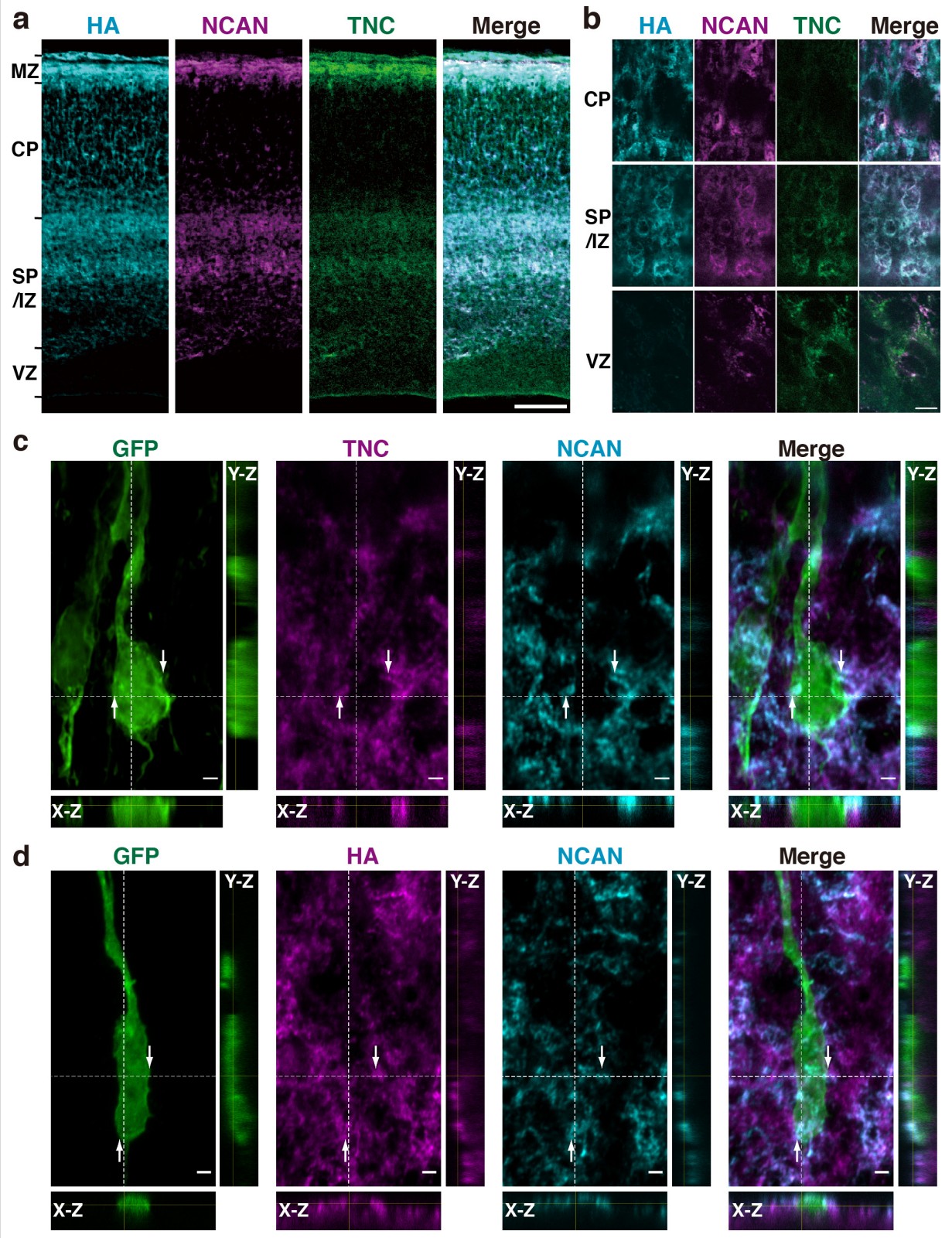

**Figure 5.** HA, NCAN, and TNC form the ternary complex in the developing cerebral cortex. (**a**) Triple staining of the E17.5 mouse cerebral cortex with HA (cyan), NCAN (magenta), and TNC (green). (**b**) High-magnification images show the co-localization of the three components in the upper part of the SP/IZ but not in the CP or VZ. (**c, d**) Localization of TNC (c, magenta), NCAN (c, d, cyan), and HA (d, magenta) around GFP-labeled bipolar neurons (green) in the upper SP/IZ at E17.5, 3 days after in utero labeling. Orthogonal views taken along the white dashed lines showed the contact between the ternary complex and the surface of bipolar neurons, as indicated by the arrows. Scale bars represent 100 μm (**a**), 5 μm (**b**), and 2 μm (**c, d**).

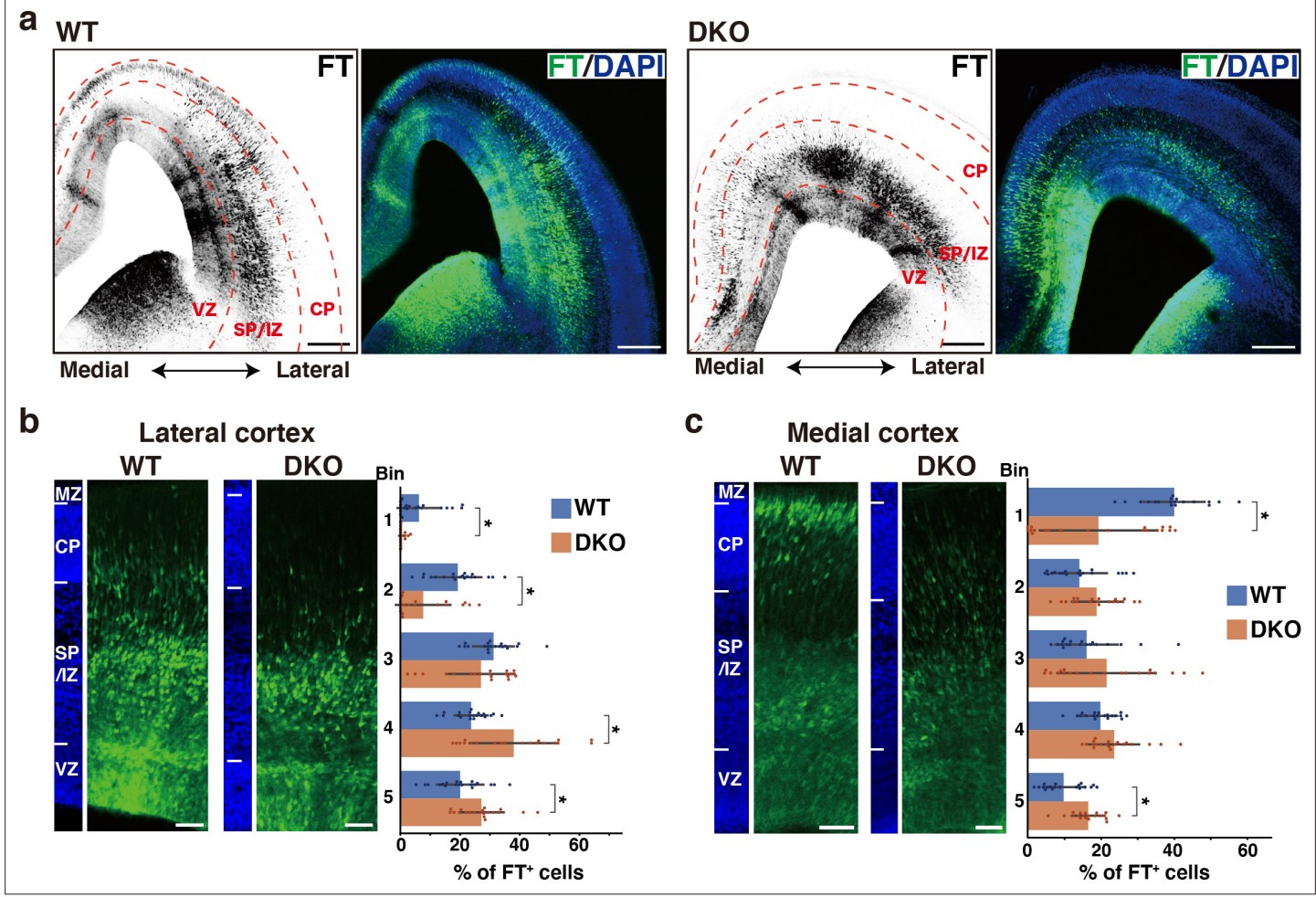

**Figure 6.** Defects in NCAN and TNC retards neuronal migration. (**a**) Low-magnification images of FT-labeled cells (black or green) in WT and DKO mouse coronal sections at E16.5. Nuclei were counterstained with DAPI (blue). (**b, c**) Radial distribution of FT-labeled cells (green) in the lateral (**b**) and medial (**c**) cortices of WT and DKO mice at E16.5. A quantitative analysis of migration profiles across the cortex is shown on the right. The cerebral cortex is divided into five equal areas (bins 1–5) from the pia to the ventricle, and the proportion of FT-labeled cells in each bin was calculated. The nuclei staining images (blue) on the left illustrate the boundaries of the cortical layers. N=19–20 mice per group. Mean ± SD; *p<0.05; Student's t-test. Scale bars represent 200 μm (**a**) and 50 μm (**b, c**).

The online version of this article includes the following source data and figure supplement(s) for figure 6:

**Source data 1.** Numerical source data for *Figure 6*.

**Figure supplement 1.** Characterization of DKO mice.

**Figure supplement 1—source data 1.** Numerical source data and original blots for *Figure 6—figure supplement 1*.

**Figure supplement 2.** Restored neuronal migration in DKO mice after 3 days of labeling.

**Figure supplement 2—source data 1.** Numerical source data for *Figure 6—figure supplement 2*.

**Figure supplement 3.** The laminar organization of the postnatal cortex in WT and DKO mice.

**Figure supplement 3—source data 1.** Numerical source data for *Figure 6—figure supplement 3*.

**Figure supplement 4.** Histochemical analysis of radial glial cells, intermediate progenitor cells, and the morphology of radial fibers in DKO mice.

**Figure supplement 4—source data 1.** Numerical source data for *Figure 6—figure supplement 4*.

We analyzed the effect of single deletion of NCAN or TNC on the ternary complex formation. In WT mice, TNC was accumulated in the SP/IZ, whereas in NCAN KO mice, the localization of TNC shifted to the lower layers (*Figure 7b*). Furthermore, TNC KO mice exhibited an attenuated accumulation of NCAN in the SP/IZ compared with WT mice (*Figure 7c*). These results suggest that TNC and NCAN are mutually required for their localization in the SP/IZ. Then, we examined whether the

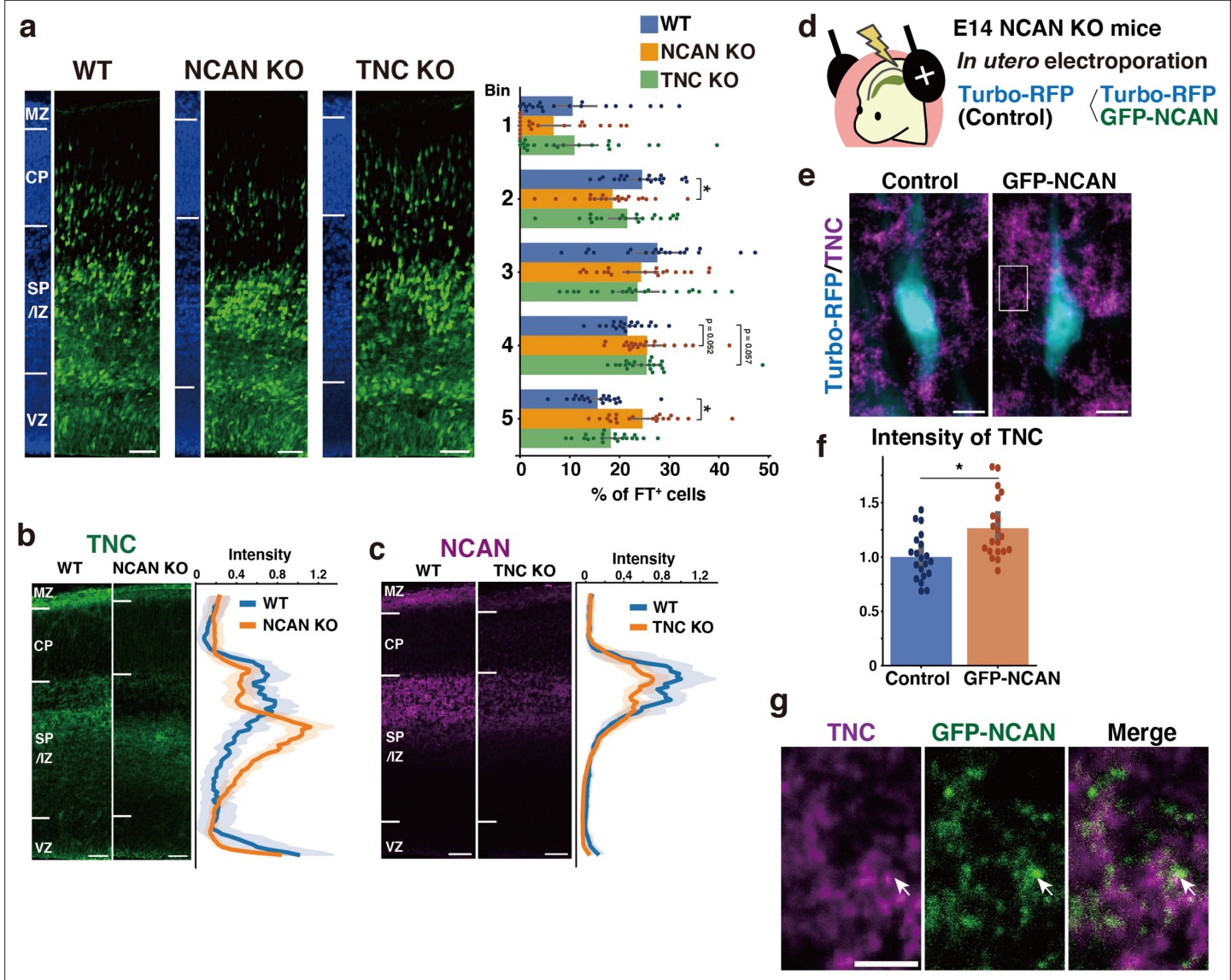

**Figure 7.** Single deletion of NCAN or TNC results in mild abnormalities in neuronal migration. (**a**) Radial distribution of FT-labeled cells (green) in the lateral cortices of WT, NCAN KO, and TNC KO mice at E16.5. A quantitative analysis of migration profiles across the cortex is shown on the right. The nuclei staining images (blue) on the left illustrate the boundaries of the cortical layers. N=20–21 mice per group. Mean ± SD; *p<0.05 vs. WT; Dunnett's test. (**b, c**) Localization of TNC in WT and NCAN KO mice at E16.5 (**b**). Localization of NCAN in WT and TNC KO mice at E16.5 (**c**). The normalized fluorescence intensity profiles of TNC and NCAN are shown on the right of the images. The maximum intensity value for WT mice was set to 1. N=8–13 mice for each group. Mean ± SD (shaded area). (**d**) In utero electroporation of Turbo-RFP alone or with GFP-fused full-length NCAN (GFP-NCAN) into the NCAN KO brain at E16.5. (**e**) Immunostaining of TNC (magenta) around Turbo-RFP-positive neurons (cyan) 2 days after electroporation. (**f**) Staining intensity of TNC around the control and GFP-NCAN-expressing neuron. N=20 images per group. Mean ± SD; *p<0.05; Student's t-test. (**g**) High magnification of the boxed region in (**e**). The arrows indicate the juxtaposed localization of TNC (magenta) and GFP-NCAN (green). Scale bars represent 50 μm (**a, b, c**), 5 μm (**e**), and 2 μm (**g**).

The online version of this article includes the following source data and figure supplement(s) for figure 7:

**Source data 1.** Numerical source data for *Figure 7*.

**Figure supplement 1.** Localization of HA in WT, DKO, NCAN KO, and TNC KO mice.

**Figure supplement 1—source data 1.** Numerical source data for *Figure 7—figure supplement 1*.

reduction in TNC accumulation observed in NCAN KO mice can be rescued by forced expression of NCAN in migrating neurons. We expressed Turbo-RFP alone or with GFP-fused full-length NCAN in NCAN KO brains by in utero electroporation at E14.5 (*Figure 7d*). Immunostaining of TNC around the transfected Turbo-RFP-positive neurons after 2 days indicated that expression of GFP-fused NCAN increased the accumulation of TNC compared to controls (*Figure 7e and f*). Higher magnification views showed that TNC was localized adjacent to GFP-fused NCAN (*Figure 7g*). Additionally, in DKO mice, the accumulation of HA, another component of the ternary complex, was also decreased in the SP/IZ (*Figure 7—figure supplement 1a*). The reduction in HA accumulation was less pronounced in single KO mice compared to DKO mice (*Figure 7—figure supplement 1b*), indicating that HA could not localize in the SP/IZ only when both NCAN and TNC were absent.

## Hyaluronidase digestion disrupts the ternary complex and inhibits neuronal migration

To further support the ternary complex formation in vivo, we degraded HA at E14.5 by intraventricular injection of hyaluronidase. The majority of HA disappeared from the cerebral cortex 2 days after the injection (*Figure 8a*). We noted that the staining intensity of NCAN and TNC significantly decreased at the SP/IZ following the hyaluronidase treatment, indicating the interaction of these molecules and the requirement of HA for the localization of NCAN and TNC. Immunoblot analysis showed that the digestion of HA reduced the amount of NCAN and TNC in the cerebral cortex by approximately half, confirming the essential role of HA in the retention of NCAN and TNC (*Figure 8b and c*). Migration assay using Flash Tag showed that in comparisons with PBS-injected controls, a hyaluronidase injection at E14.5 inhibited neuronal migration into the CP in the lateral and medial cortices (*Figure 8d and e*). However, the impact of hyaluronidase treatment on neuronal migration was not as prominent as observed in DKO mice. This result could be attributed to the fact that approximately half of NCAN and TNC remained in the tissue after hyaluronidase digestion.

## DKO mice show delayed multipolar-to-bipolar transition during radial migration

To gain insights into the mechanisms underlying migration defects in DKO mice, we investigated whether the loss of NCAN and TNC influenced the multipolar-to-bipolar transition. Fifty-three hours after in utero electroporation with a cytosolic form of mCherry, we evaluated neuronal morphology based on the following criteria: (1) the ratio of the length-to-width of labeled neurons as an indicator of cell shape; (2) the angle between major neurites and the ventricle surface; (3) the percentage of neurons with a bipolar morphology, defined as cells with a length-to-width ratio greater than three and an orientation angle greater than 70° (*Figure 9a*). In utero electroporation-mediated labeling also showed delayed neuronal migration in DKO mice, similar to Flask Tag labeling (*Figure 9b*). Morphological analysis revealed a significantly lower length-to-width ratio of neurons in DKO mice than in WT mice (*Figure 9c and d*). This result indicated that control neurons had an elongated shape, whereas neurons in DKO mice retained a rounded shape. Many neurons radially extended their major neurites toward the pia surface (between 70° and 90° relative to the ventricle surface) in WT mice, which is characteristic of bipolar neurons (*Figure 9e*). Compared with control neurons, the major neurites of DKO neurons were not directed toward the pia surface. Consistent with these results, we observed a significantly smaller proportion of bipolar neurons in DKO mice than in WT mice (*Figure 9f*). Therefore, retarded neuronal migration in DKO mice was associated with defects in the multipolar-to-bipolar transition.

## Morphological maturation of cortical neurons by TNC and NCAN

Next, we analyzed the effects of each component of the ternary complex on the morphological maturation of cortical neurons in vitro. We labeled progenitor cells with GFP at E14.5 by in utero electroporation and cultured dissociated neurons prepared at E15.5 on a glass plate coated with HA, NCAN, or TNC (*Figure 10a*). GFP-positive neurons on the control substrate, poly-L-ornithine (PLO), exhibited a typical morphological transition from stages 1–3 as follows (*Figure 10b*): Neurons initially have immature processes (stage 1) and then begin to extend multiple neurites of approximately equal lengths (stage 2); One of the neurites rapidly elongates to become the axon (stage 3), corresponding to the multipolar-to-bipolar transition in vivo. After 2 days of plating, culturing neurons on the TNC-coated

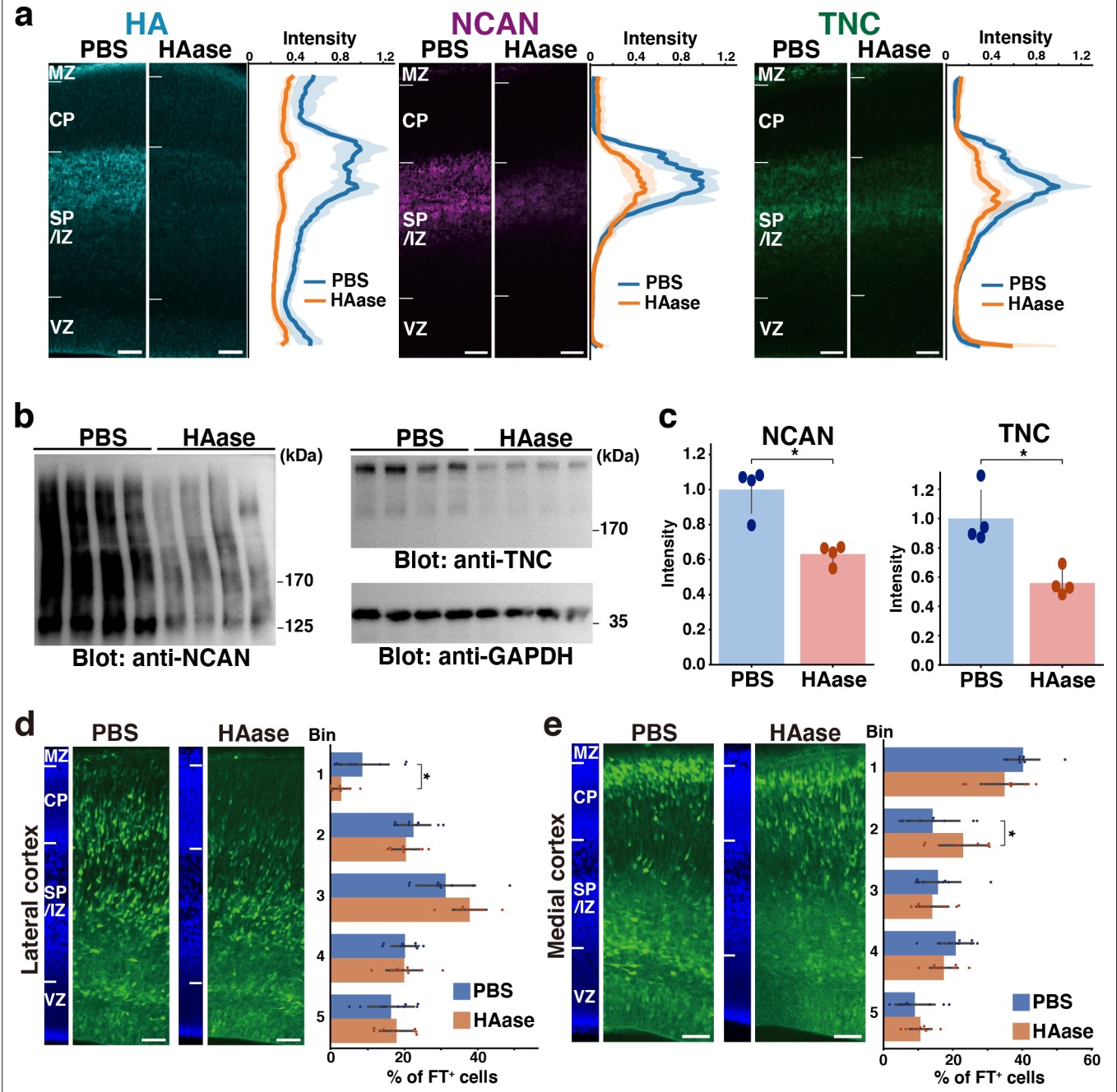

**Figure 8.** Transient disruption of the ternary complex by hyaluronidase injection. (**a**) Localization of HA, NCAN, and TNC in the cerebral cortex 2 days after intraventricular injection of PBS or hyaluronidase (HAase) at E14.5. The normalized fluorescence intensity profiles of HA, NCAN, and TNC are shown on the right of each image. The maximum intensity value for PBS-injected mice was set to 1. N=5 mice for each group. Mean ± SD (shaded area). (**b**) Immunoblot analysis of NCAN and TNC in cerebral cortex lysates 2 days after intraventricular injection of PBS or HAase. The broadening of the NCAN band is due to the absence of chondroitinase ABC digestion. (**c**) The NCAN (left) and TNC (right) amounts were represented relative to the PBS-injected group. N=4 mice per group. Mean ± SD; *p<0.01; Student's t-test. (**d, e**) Distribution of FT-labeled cells (green) in the lateral (**d**) and medial (**e**) cortices 2 days after injection of PBS or HAase at E14.5. N=9–10 mice per group. Mean ± SD; *p<0.05; Student's t-test. Scale bars represent 50 μm.

The online version of this article includes the following source data for figure 8:

**Source data 1.** Numerical source data and original blots for *Figure 8*.

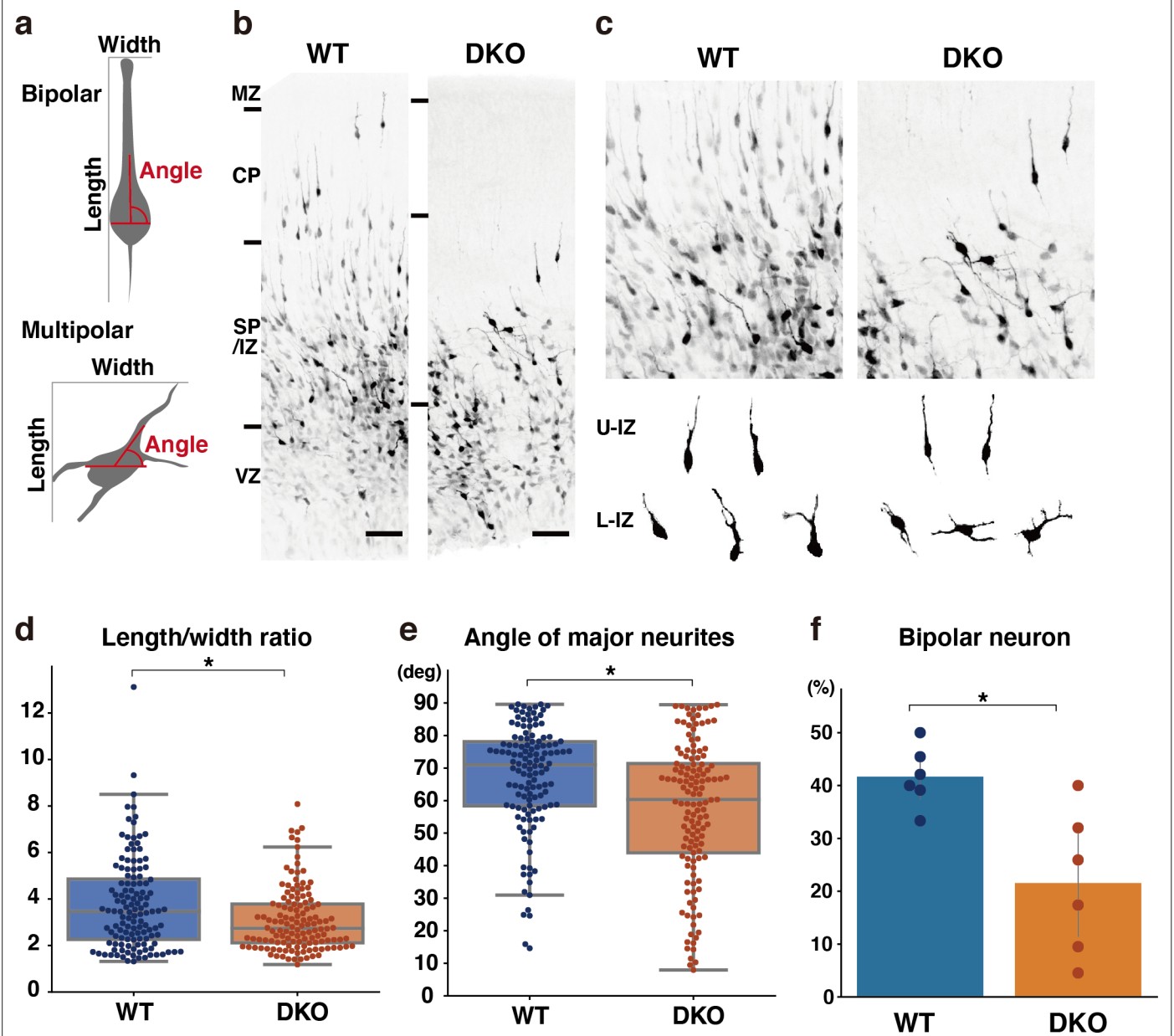

**Figure 9.** Delayed multipolar-to-bipolar transition in DKO mice. (**a**) Schematic of the morphological analysis. (**b, c**) Migration of mCherry-labeled cells in the WT and DKO cerebral cortices 53 hr after in utero labeling (**b**). High-magnification images of mCherry-labeled cells in the IZ (**c**). The images below show the morphology of neurons in the upper IZ (U–IZ) and lower IZ (L–IZ). (**d, e**) The length-to-width ratio (**e**) and the major neurite angle (**f**) of mCherry-positive neurons in WT and DKO mice. N=128–134 cells from 6 mice per group. *p<0.05; Student's t-test. (**f**) The proportion of bipolar neurons among mCherry-labeled neurons in the WT and DKO cortices. N=6 mice per group. Mean ± SD; *p<0.05; Student's t-test. Scale bars represent 50 µm (**b**).

The online version of this article includes the following source data for figure 9:

**Source data 1.** Numerical source data for *Figure 9*.

plates significantly accelerated neurite outgrowth (*Figure 10c*). The length of the longest neurite notably increased when the glass plate was coated with 10 µg/mL of TNC (*Figure 10d*). In contrast, coating with 10 µg/mL of HA or NCAN did not significantly affect neurite elongation. Plating neurons on TNC also increased the total length and number of neurites (*Figure 10e and f*). This effect of TNC on neurite outgrowth was also evident after 3 days of plating (*Figure 10—figure supplement 1a–d*). The impact of TNC was further confirmed when the neurites of all neurons were stained with anti-Tuj-1 antibody instead of labeling a small population of neurons through in utero electroporation

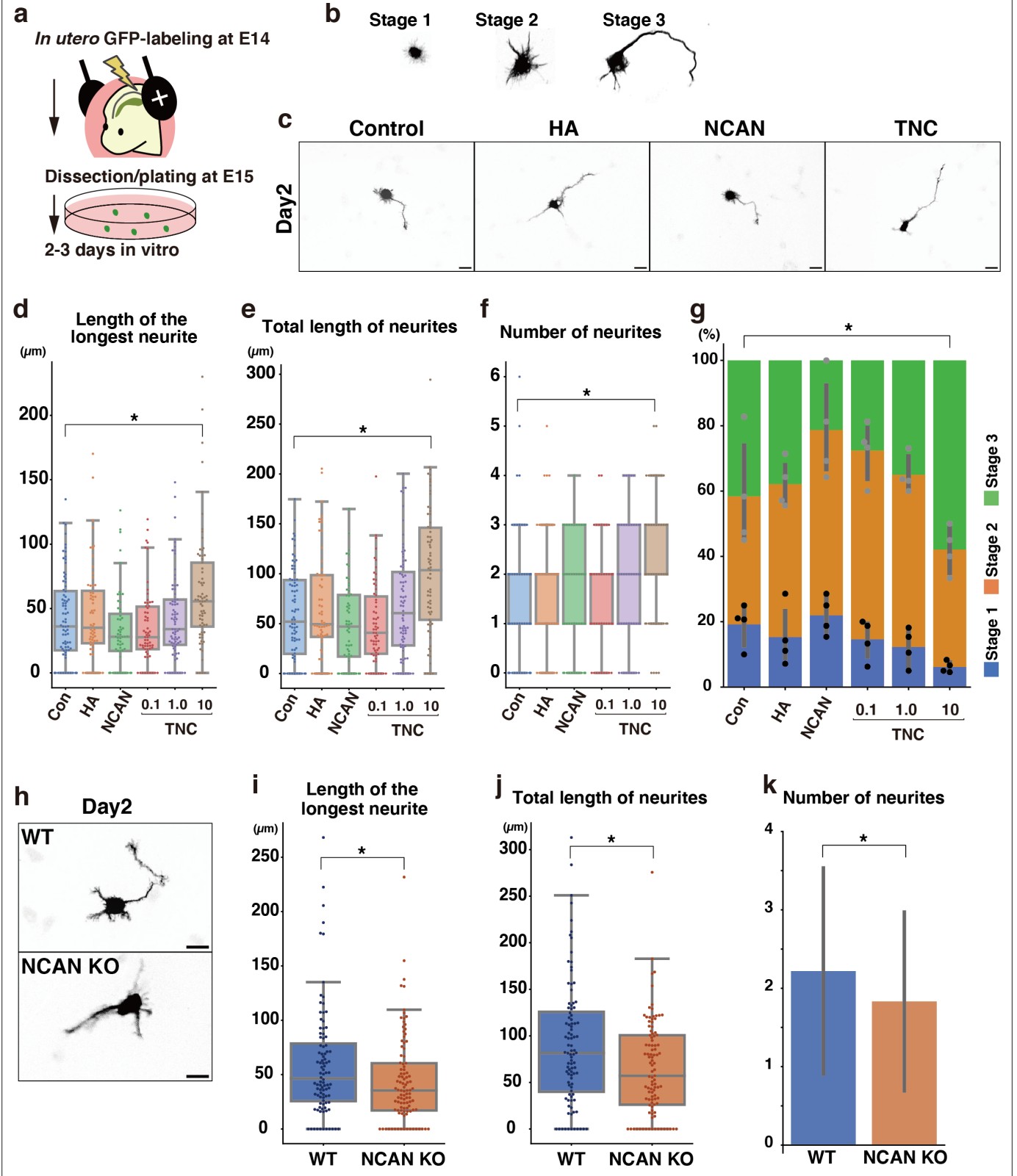

**Figure 10.** Morphological maturation of cortical neurons by TNC and NCAN. (**a**) Experimental model for in utero cell labeling and the primary neuronal culture. (**b**) Morphological stages of primary cultured cortical neurons. (**c**) Representative images of GFP-labeled neurons cultured for 2 days on cover glasses coated with the control substrate, poly-L-ornithine (POL), and 10 µg/mL of TNC, HA, and NCAN. (**d-f**) The length of the longest neurite (**d**), total length of neurites (**e**), and number of neurites (**f**) of neurons cultured on the indicated substrate for 2 days. N=55–80 cells per condition. *p<0.05 vs

*Figure 10 continued on next page*

*Figure 10 continued*

Control (Con); Dunnett's test. (**g**) The percentage of neurons with each morphological stage after culturing on the indicated substrate for 2 days. N=4 wells per condition. Mean ± SD. *p<0.05 vs Con for stage 1; Dunnett's test. (**h**) Representative images of GFP-labeled neurons derived from WT and NCAN KO mice cultured for 2 days on cover glasses coated with POL. (**i–k**) The length of the longest neurite (**i**), total length of neurites (**j**), and number of neurites (**k**) of WT and NCAN KO neurons cultured for 2 days. N=96–101 cells per condition. *p<0.05; Student's t-test. Scale bars represent 20 μm (**c**, **h**).

The online version of this article includes the following source data and figure supplement(s) for figure 10:

**Source data 1.** Numerical source data for *Figure 10*.

**Figure supplement 1.** Morphological analysis of cortical neurons cultured for 3 days on cover glasses coated with HA, NCAN, and TNC.

**Figure supplement 1—source data 1.** Numerical source data for *Figure 10—figure supplement 1*.

**Figure supplement 2.** Neurite outgrowth analysis with anti-Tuj-1 antibody staining.

**Figure supplement 2—source data 1.** Numerical source data for *Figure 10—figure supplement 2*.

**Figure supplement 3.** Neurite outgrowth analysis of WT and NCAN KO neurons with anti-Tuj-1 antibody staining.

**Figure supplement 3—source data 1.** Numerical source data for *Figure 10—figure supplement 3*.

---

(*Figure 10—figure supplement 2a–e*). Closer observations found that TNC enhances the morphological transition of cortical neurons from stages 1 to 3 in a dose-dependent manner (*Figure 10g*), suggesting that TNC stimulates the multipolar-to-bipolar transition.

However, these findings did not rule out the potential involvement of NCAN and HA in the morphological transition since cortical neurons themselves are capable of secreting NCAN and HA in culture (*Figures 1j and 2d*). Indeed, in a previous study, we reported that pharmacological inhibition of HA synthesis led to the suppression of neurite outgrowth (*Takechi et al., 2020*). To further investigate the possible role of NCAN in neurite outgrowth, we cultured cortical neurons derived from WT and NCAN KO embryos (*Figure 10h*). After 2 and 3 days of plating, NCAN KO neurons exhibited a significantly suppressed elongation of the longest neurite compared to WT neurons (*Figure 10h and i*, *Figure 10—figure supplement 3a, b*). Additionally, the deletion of NCAN reduced the total length and number of neurites (*Figure 10j and k*, *Figure 10—figure supplement 3c, d*). These results indicated that TNC and NCAN play critical roles in the morphological maturation of cortical neurons.

## Discussion

The multipolar-to-bipolar transition of developing cortical neurons is essential for radial migration and neuronal polarity. Defects in the multipolar-to-bipolar transition lead to neuronal mispositioning, laminar structure disorganization, and cortical development malformation (*Castello and Gleeson, 2021*; *Fernández et al., 2016*). Therefore, the timing and location of the multipolar-to-bipolar transition need to be tightly controlled. The present results propose that the assembly of ECM molecules in the SP/IZ provides an extracellular positional cue for migrating neurons that specify the site of the multipolar-to-bipolar transition. Polarity formation in vivo occurs in a microenvironment or niche created in the SP/IZ. A previous study found that pre-existing efferent axon tracts in the SP/IZ interacted with migrating multipolar neurons to promote the formation of axons (*Namba et al., 2014*). Secreted soluble factors also play significant roles in this process. A gradient of semaphorin 3 A in the developing cerebral cortex regulates neuronal migration and polarity by attracting apical dendrites and repulsing axons (*Chen et al., 2008*; *Polleux et al., 2000*). Transforming growth factor-β initiates signaling pathways that specify axons (*Yi et al., 2010*). Reelin, a glycoprotein secreted by Cajal-Retzius cells in the MZ, is essential for completing neuronal migration and cortical lamination (*D'Arcangelo et al., 1995*; *Jossin and Cooper, 2011*). Therefore, a combination of ECM molecules, secreted factors, and neighboring cells regulates neuronal polarity during cortical development.

Radial glial cells play a dual function during development: producing neurons and serving as scaffolds for neuronal migration. The present study discovered another role for radial glial cells: they secrete and deposit TNC to stimulate the morphological transition of migrating neurons in the SP/IZ. In gyrencephalic species, such as humans, basal/outer radial glial cells residing in the outer sub-VZ generate the majority of cortical neurons and are responsible for the expansion of the cerebral cortex (*Kriegstein and Alvarez-Buylla, 2009*; *Taverna et al., 2014*). Basal/outer radial glial cells are characterized by the high gene expression of several ECM molecules, including TNC (*Fietz et al., 2012*;

*Pollen et al., 2015*), indicating that basal/outer radial glial cells create a microenvironment that promotes neuronal migration. In addition, HA-based ECM affects the cortical folding of the developing human neocortex (*Long et al., 2018*), suggesting that the evolutionarily conserved ECM structure contributes to neocortex expansion in gyrencephalic species.

The tenascin gene family in vertebrates comprises four closely related members. They share structural motifs: a cysteine-rich region at the N-terminus followed by EGF-like repeats, FNIIIs, and a fibrinogen-like globe at the C-terminus. TNC monomers assemble into hexamers at the N-terminal region. Distinct domains of TNC are involved in cell adhesion, cell repulsion, and neurite outgrowth promotion (*Andrews et al., 2009*; *Faissner et al., 2017*; *Götz et al., 1997*; *Lochter et al., 1991*; *Tucker and Chiquet-Ehrismann, 2015*). For example, alternatively spliced FNIIIs stimulate neurite outgrowth by binding to neuronal surface contactin-1 in primary cultured neurons (*Rigato et al., 2002*). Several integrins have been identified as receptors for TNC, and through this interaction, TNC influences cell adhesion, migration, and proliferation (*Tucker and Chiquet-Ehrismann, 2015*). However, the mechanisms by which TNC secreted from radial glial cells are presented to neurons in vivo remain unclear. Our analysis revealed that binding to HA via NCAN was required for the deposition of TNC in the SP/IZ. Further studies are needed to identify receptors for TNC on migrating neurons and its downstream signaling pathway that transmits information from the extracellular environment to the inside of the cell.

There is a transition from embryonic- to adult-type ECM during brain development (*Miyata et al., 2012*; *Zimmermann and Dours-Zimmermann, 2008*). The expression of NCAN and TNC is high in the embryonic brain, whereas ACAN, brevican, and TNR increase during late postnatal development. In the adult brain, ACAN cross-links HA and TNR and forms macromolecular aggregates around subpopulations of neurons (*Fawcett et al., 2022*; *Miyata and Kitagawa, 2017*). This condensed ECM structure, called perineuronal nets, is formed at the late postnatal stages and plays pivotal roles in neural plasticity, memory consolidation, and circuit maturation (*Fawcett et al., 2022*). The ternary complex of ACAN, HA, and TNR physically stabilizes existing synapses and inhibits new synapse formation. In perineuronal nets, high-molecular-weight HA serves as a structural scaffold, facilitating the accumulation of ECM around subpopulations of neurons in the adult brain. The present study showed that a structurally similar ternary complex of NCAN, HA, and TNC was formed in the SP/IZ of the embryonic cortex but exerted distinct functions in cortical development. Previous studies have reported an upregulation of NCAN and TNC in reactive astrocytes, indicating the potential formation of the ternary complex of NCAN, TNC, and HA in the adult brain in response to injury (*Deller et al., 1997*; *Haas et al., 1999*).

The enzymatic digestion of HA impedes the accumulation of NCAN and TNC within the SP/IZ, supporting that HA functions as a scaffold for ECM in the developing brain. Alternatively, HA may transmit extracellular signals to developing neurons through cell surface HA receptors such as CD44 (*Skupien et al., 2014*). Although the interaction of NCAN with HA and TNC was previously reported in vitro, the present results revealed the formation of the ternary complex in vivo for the first time. Full-length NCAN is detected during early development but not in adulthood, indicating that the proteolytic cleavage of NCAN affects the assembly of the ternary complex. Screening using peptide library mixtures revealed that matrix metalloprotease-2 cleaved the central region of NCAN in vitro (*Turk et al., 2001*). The temporal control of matrix metalloproteases may affect neuronal migration via the assembly and disassembly of the ECM.

We observed that single deletion of NCAN leads to mild abnormalities in neuronal migration. The loss of NCAN may interfere with TNC localization and thus delay neuronal migration. Alternatively, the binding of NCAN to cell adhesion molecules on the neuronal surfaces may affect neurite outgrowth (*Friedlander et al., 1994*; *Rauch et al., 2001*). Genome-wide association studies identified a genetic variation in NCAN as a risk factor for bipolar disorder and schizophrenia. However, the relationship between the delayed neuronal migration observed in NCAN KO mice and the onset of neuropsychiatric disorders remains uncertain. In adult NCAN KO mice, there are no apparent deficits in brain anatomy, morphology, or ultrastructure (*Zhou et al., 2001*), which is consistent with our results that the delay in neuronal migration in DKO mice is transient. A recent study reported that the C-terminal portion of NCAN promotes synapse formation during postnatal development, which may be more relevant to neuropsychiatric disorders (*Irala et al., 2023*). These findings suggest that NCAN has different functions in the embryonic and adult brain.

In conclusion, the present study revealed that the assembly of NCAN, HA, and TNC forms the microenvironment in the SP/IZ, in which the multipolar-to-bipolar transition occurs. We uncovered that enzymatic and genetic disruption of the ternary complex impairs radial migration by inhibiting the multipolar-to-bipolar transition. Our findings shed light on the collaborative role of ECM molecules derived from neurons and radial glial cells in cortical development.

# Materials and methods

### Key resources table

| Reagent type (species) or resource | Designation | Source or reference | Identifiers | Additional information |
|---|---|---|---|---|
| Strain, strain background (*Mus musculus*) | ICR mice | Japan SLC | RRID:MGI:5462094 | |
| Strain, strain background (*M. musculus*) | C57BL/6 N mice | Japan SLC | RRID:MGI:5295404 | |
| Strain, strain background (*M. musculus*) | B6.Cg-Tnc<tm1Sia>/Rbrc | RIKEN Bioresource Center | RBRC00169 | |
| Strain, strain background (*M. musculus*) | DKO mice for TNC and NCAN | This paper | N/A | Mice deficient for TNC and NCAN |
| Cell line (*Homo sapiens*) | HEK293 | Riken Cell Bank | RCB1637 RRID:CVCL_0045 | Verified by the Riken Cell Bank and tested negative for mycoplasma |
| Cell line (*H. sapiens*) | HEK293T | Riken Cell Bank | RCB2202 RRID:CVCL_0063 | Verified by the Riken Cell Bank and tested negative for mycoplasma |
| Antibody | Anti-tenascin C Rat IgG2A (Clone # 578) | R&D | MAB2138, RRID:AB_2203818 | IF(1:400), WB (1:3000) |
| Antibody | Anti-β-Tubulin (Tuj-1) Mouse IgG1(clone TUB 2.1) | Sigma | T4026 RRID:AB_477577 | IF(1:1000) |
| Antibody | Anti-Neurocan Sheep IgG | R&D | AF5800 RRID:AB_2149717 | IF(1:400), WB (1:3000) |
| Antibody | Anti-GFP Alexa Fluor 488 conjugate Rabbit IgG | Invitrogen | A21311 RRID:AB_221477 | IF(1:1000) |
| Antibody | Anti-Nestin Mouse IgG | Millipore | MAB5326 RRID:AB_94911 | IF(1:400) |
| Antibody | Anti-Pax6 Rabbit IgG | Fujifilm | 015–27293 | IF(1:400) |
| Antibody | Anti-EOMES (Tbr2) Rat IgG2A | Invitrogen | 14-4875-82 RRID:AB_11042577 | IF(1:400) |
| Antibody | Anti-NeuN Rabbit IgG | Proteintech | 26975–1-AP RRID:AB_2880708 | IF(1:1000) |
| Antibody | Anti-Ctip2 Rat IgG2A | abcam | Ab18465 RRID:AB_2064130 | IF(1:400) |
| Antibody | Anti-GAPDH Mouse IgG1(Clone # 5A12) | Wako | 016–25523 RRID:AB_2814991 | WB (1:10000) |
| Antibody | Anti-Chondroitine-4-Sulfate Mouse IgG1 (Clone # BE-123) | Millipore | MAB2030 RRID:AB_11213679 | WB (1:50000) |
| Antibody | Alexa Fluor 488 donkey anti-Mouse (H+L) | Wako | 715-545-151 RRID:AB_2341099 | IF(1:400) |
| Antibody | Alexa Fluor 594 goat anti-Rat IgG (H+L) | Thermo Fisher | A-11007 RRID:AB_10561522 | IF(1:400) |
| Antibody | Alexa Fluor 647 goat anti-Rat IgG (H+L) | Thermo Fisher | A-21247 RRID:AB_141778 | IF(1:400) |
| Antibody | Alexa Fluor 647 donkey anti-Sheep IgG (H+L) | Thermo Fisher | A-21448 RRID:AB_2535865 | IF(1:400) |
| Antibody | Mouse IgG HRP-conjugated Antibody | MBL | PM009-7 | WB (1:2500) |
| Antibody | Sheep IgG HRP-conjugated Antibody | R&D | HAF016 RRID:AB_562591 | WB (1:2500) |

*Continued on next page*

*Continued*

| Reagent type (species) or resource | Designation | Source or reference | Identifiers | Additional information |
|---|---|---|---|---|
| Antibody | Rat IgG HRP-conjugated Antibody | Cell Signaling Technology | 7077 S RRID:AB_10694715 | WB (1:2500) |
| Antibody | Rabbit IgG HRP-conjugated Antibody | Cell Signaling Technology | 7074 S RRID:AB_2099233 | WB (1:2500) |
| Recombinant DNA reagent | pCAG-GFP (plasmid) | Addgene | Plasmid #11150 | |
| Recombinant DNA reagent | pCAG-mGFP (plasmid) | Addgene | Plasmid #14757 | |
| Recombinant DNA reagent | pCAG-mCherry (plasmid) | This paper | N/A | Vector backbone: pCAG Gene/Insert name: mCherry |
| Recombinant DNA reagent | pCAGGS-TurboRFP (plasmid) | This paper | N/A | Vector backbone: pCAGGS Gene/Insert name: TurboRFP |
| Recombinant DNA reagent | pCAG-Full length NCAN-GFP (plasmid) | This paper | N/A | Vector backbone: pCAG Gene/Insert name: GFP-fused full length NCAN |
| Recombinant DNA reagent | pCAG-N half NCAN-GFP (plasmid) | This paper | N/A | Vector backbone: pCAG Gene/Insert name: GFP-fused N half of NCAN |
| Recombinant DNA reagent | pCAG-C half NCAN-GFP (plasmid) | This paper | N/A | Vector backbone: pCAG Gene/Insert name: GFP-fused C half of NCAN |
| Sequence-based reagent | ISH TNC probe f | This paper | N/A | CGGAATTCATCTTTGC AGAGAAAGGACAGC |
| Sequence-based reagent | ISH TNC probe r | This paper | N/A | GCTCTAGACTGTGTCCT TGTCATAGGTGGA |
| Sequence-based reagent | ISH NCAN probe f | This paper | N/A | GCGAATTCAGAATGCCTC TCTTGTTGGTG |
| Sequence-based reagent | ISH NCAN probe r | This paper | N/A | GCTCTAGACTACAATAGT GAGTTCGAGGCC |
| Sequence-based reagent | crRNA, Mm.Cas9.NCAN.1.AA | IDT | REF #101658344 | ACCUUAGUCCACUUGAU CCGGUUUUAGAGCUAUGCU |
| Sequence-based reagent | qPCR for Ncan, forward | This paper | N/A | CCCTGCTTCTTTACCCTGCA |
| Sequence-based reagent | qPCR for Ncan, reverse, | This paper | N/A | CGTTGTCTTTGGCCACCAAG-3' |
| Sequence-based reagent | qPCR for Tnc, forward | This paper | N/A | ACCATGGGTACAGGCTGTTG |
| Sequence-based reagent | qPCR for Tnc, reverse | This paper | N/A | CCTTTCCAGCCTGGTTCACA |
| Sequence-based reagent | qPCR for Gapdh, forward | This paper | N/A | GACTTCAACAGCAACTCCCAC |
| Sequence-based reagent | qPCR for Gapdh, reverse | This paper | N/A | TCCACCACCCTGTTGCTGTA |
| Peptide, recombinant protein | Alt-R S.p. Cas9 Nuclease V3 | IDT | Catalog #1081058 | |
| Peptide, recombinant protein | Trypsin (Sequencing Grade) | Promega | V511C | |
| Peptide, recombinant protein | HRP-conjugated streptavidin | Wako | 190–17441 | |
| Peptide, recombinant protein | Hyaluronidase from streptomyces hyalurolyticus | Sigma | H1136 | |
| Peptide, recombinant protein | Recombinant Human Tenascin C Protein | R&D | 3358-TC | |

*Continued on next page*

*Continued*

| Reagent type (species) or resource | Designation | Source or reference | Identifiers | Additional information |
|---|---|---|---|---|
| Peptide, recombinant protein | Recombinant Human Neurocan Protein | R&D | 6508-NC-050 | |
| Peptide, recombinant protein | Chondroitinase ABC Protease Free | Seikagaku Corporation | 100332 | |
| Commercial assay or kit | BCA assay kit | Thermo Fisher | 23227 | |
| Commercial assay or kit | RNeasy Mini Kit | Qiagen | 74104 | |
| Commercial assay or kit | PowerUp SYBR Green Master Mix | Thermo Fisher | A25741 | |
| Commercial assay or kit | DIG RNA Labeling Kit (SP6/T7) | Roche | 11175025910 | |
| Commercial assay or kit | NEBuilder HiFi DNA Assembly Master Mix | New England Biolabs | E2621 | |
| Commercial assay or kit | In-Fusion HD Cloning Kit | Takara | 639648 | |
| Commercial assay or kit | KOD -Plus- Mutagenesis Kit | Toyobo | SMK-101 | |
| Software, algorithm | MASCOT server | Matrix science | https://www.matrixscience.com/server.html RRID:SCR_014322 | |
| Software, algorithm | ImageJ | *Schneider et al., 2012* | https://imagej.nih.gov/ij/ | |
| Other | DAPI solution | DOJINDO | Lot.PF082 340–07971 | |
| Other | N-2 Supplement | Thermo Fisher | 17502–048 | |
| Other | B-27 Supplement | Thermo Fisher | 17504–044 | |
| Other | Neurobasal Medium | Thermo Fisher | 21103–049 | |
| Other | Penicillin-Streptomycin | Thermo Fisher | 15070–063 | |
| Other | HBSS (X10) | Thermo Fisher | 14185–052 | |
| Other | HEPES (1 M) | Thermo Fisher | 15630–080 | |
| Other | Papain | Worthington Biochemical Corporation | LK003178 | |
| Other | Fast Green | Wako | 061–00031 | |
| Other | Protease Inhibitor Cocktail | Sigma | P8340-1ML | |
| Other | Streptavidin Magnetic Beads | Thermo Fisher | 88817 | |
| Other | Poly-L-Ornithine Solution | Wako | 163–27421 | |
| Other | Skim Milk Powder | Wako | 190–12865 | |
| Other | Immobilon Transfer Membrane PVDF | Millipore | IPVH00010 | |
| Other | Immobilon Western Chemiluminescent | Millipore | WBKLS0500 | |
| Other | Sodium Hyaluronate (M2) (Mw:600,000~1,120,000) | PG Research | NaHA-M2 | |
| Other | Biotinylated Sodium Hyaluronate (M1) (Mw:600,000~1,120,000) | PG Research | BHHA-M1 | |
| Other | Carboxyfluorescein diacetate succinimidyl ester | DOJINDO | 341–07401 | |
| Other | GFP-Trap-Agarose | Chromotek | gta-20 | |
| Other | ELISA 96 well plate | IWAKI | 3801–096 | |
| Other | BLOCK ACE Powder | KAC | UKB80 | |
| Other | ELISA POD Substrate TMB Solution | Nacalai | 05299–54 | |

*Continued on next page*

*Continued*

| Reagent type (species) or resource | Designation | Source or reference | Identifiers | Additional information |
|---|---|---|---|---|
| Other | Hyaluronan Binding Protein (HABP) | Cosmo-bio | BC40 | |
| Other | Hyaluronic Acid Sodium Salt | Wako | 083–10341 | |
| Other | Streptavidin, Alexa Fluor 488 conjugate | Thermo Fisher | S-11223 | |
| Other | Biotinylated Hyaluronan Binding Protein | Hokudo | BC41 | |
| Other | SuperScript III reverse transcriptase | Thermo Fisher | 18080044 | |
| Other | OCT Compound | Sakura Finetek | 45833 | |
| Other | Anion exchange chromatography resin | Tosoh | TOYOPEARL DEAE-650M | |
| Other | Heparin-agarose | Sigma | H6508 | |
| Other | Ultrafiltration filter | Amicon | YM-10 | |
| Other | α-cyano-4-hydroxycinnamic acid | Shimadzu | 70990 | |
| Other | Lipofectamine 3000 Reagent | Thermo Fisher | L3000008 | |
| Other | Electroporator | NEPA GENE | NEPA21 | |
| Other | 5 mm-diameter tweezers-type disc electrodes | NEPA GENE | CUY650P5 | |
| Other | 2.5 mm x 4 mm tweezers-type disc electrodes | NEPA GENE | CUY652P2.5X4 | |
| Other | Vibratome | Leica | VT1200S | |
| Other | LuminoGraph | ATTO | WSE-6100H-ACP | |
| Other | Confocal laser scanning microscope | Nikon | AX | |
| Other | Confocal laser scanning microscope | Carl Zeiss | LSM 710 NLO | |
| Other | Fluorescence stereomicroscope | Leica | M165FC | |
| Other | Animal anesthetizer | Muromachi | MK-AT210 | |
| Other | StepOne Real-Time PCR System | Thermo Fisher | 4376374 | |
| Other | Cryostat | Leica | CM3050 S | |
| Other | Direct nanoLC/MALDI fraction system | KYA technologies | DiNa-MaP | |
| Other | MALDI mass spectrometer | SCIEX | TOF/TOF 5800 | |
| Other | LC-MS/MS System | SCIEX | QTRAP 5500 | |
| Other | Fluorescence-activated cell sorter | Bay bioscience | JSAN | |

## Mice used in the present study

All animal experimental studies were conducted with the approval of the Committee on Animal Experiments of the Graduate School of Bioagricultural Sciences, Nagoya University, and Tokyo University of Agriculture and Technology (Approval Code: R02-111, R03-103, R04-220). Pregnant ICR (RRID:MGI:5462094) and C57BL/6NCrSlc (RRID:MGI:5295404) mice were purchased from Japan SLC (Shizuoka, Japan). TNC KO mice on a C57BL/6NJcl background were obtained from the RIKEN Bioresource Center (BRC No. RBRC00169; *Nakao et al., 1998*). NCAN KO mice were generated using CRISPR-Cas9-based improved genome editing via oviductal nucleic acids delivery (i-GONAD; *Gurumurthy et al., 2019*; *Ohtsuka et al., 2018*). A guide RNA targeting the mouse *Ncan* loci (Mm. Cas9.NCAN.1.AA: 5′-ACCTTAGTCCACTTGATCCGAGG-3′) was chosen from Predesigned Alt-R CRISPR-Cas9 guide RNA on the Integrated DNA Technologies (IDT) website (https://sg.idtdna.com). Synthetic crRNAs, tracrRNA, and Cas9 Nuclease V3 were purchased from IDT. Surgical procedures for i-GONAD were performed under deep anesthesia at approximately 3 pm on the day on which a vaginal plug was detected. A mixture of 30 μM crRNA/tracrRNA complex and 1 mg/mL Cas9 Nuclease V3 was prepared in PBS. Approximately 2.5 μL of the mixture was injected into the oviduct upstream of the ampulla using a glass micropipette. The oviduct was clasped with a paired electrode (CUY652P2.5X4, NEPA Gene). Square electric pulses (poring pulse: 50 V, 5 ms pulse, 50 ms pulse interval, 3 pulses,

10% decay, ±pulse orientation, and transfer pulse: 10 V, 50 ms pulse, 50 ms pulse interval, 3 pulses, 40% decay, ±pulse orientation) were passed using an electroporator (NEPA21, NEPA Gene). Offspring were genotyped by DNA sequencing to identify founder animals. After first mating with TNC KO mice, the F1 generation was sequenced and further crossed to TNC KO mice to obtain DKO mice.

## Cell lines used in the present study

The HEK293 (RCB1637) and HEK293T (RCB2202) cell lines were obtained from the Riken Cell Bank (Tsukuba, Japan) and used for protein pull-down experiments. All cell lines were verified by the Riken Cell Bank and tested negative for mycoplasma.

## Immunoblotting

Mouse brains were homogenized in PBS containing 1% Triton X-100 and protease inhibitor cocktail (Sigma) and incubated on ice for 30 min. After centrifugation at 15,000 × $g$ at 4 °C for 30 min, the protein concentrations of supernatants (brain lysates) were assessed using a BCA assay kit (Thermo Fisher). Regarding chondroitinase ABC digestion, the brain lysate (70 µg as protein) was digested with 5 milliunits of protease-free chondroitinase ABC (Sikagaku Corp) at 37 °C for 1 hr. Digested and undigested lysates were denatured with 2% sodium dodecyl sulfate (SDS) and 5% mercaptoethanol at 95 °C for 5 min. Proteins were separated using 6 or 10% SDS-polyacrylamide gel electrophoresis (PAGE) and transferred to polyvinylidene difluoride membranes. After blocking with 2% skim milk in PBS containing 0.1% Tween-20 (PBST), the membranes were incubated with an antibody recognizing the neoepitope of CSPGs (mouse, 1:50,000, Millipore), anti-NCAN (sheep, 1:3000, R&D), anti-glyceraldehyde 3-phosphate dehydrogenase (GAPDH; mouse, 1:10,000, Wako), anti-TNC (rat, 1:3000, R&D), and anti-GFP (rabbit, 1:3000, MBL) at 4 °C overnight. After washing, the membranes were incubated with appropriate horseradish peroxidase (HRP)-conjugated secondary antibodies (1:2500) at room temperature for 1 hr. Signals were detected with Immobilon Western Chemiluminescent HRP Substrate (Millipore) and imaged on a LuminoGraph I (ATTO).

## Histochemistry

Mouse embryos were transcardially perfused with PBS, followed by 4% paraformaldehyde in PBS. Brains were removed and post-fixed overnight with 4% paraformaldehyde in PBS. Coronal sections (thickness of 80 µm) were cut with the vibratome VT1200 (Leica). Sections were permeabilized with 0.25% Triton X-100 in PBS and then blocked with PBST containing 2% bovine serum albumin (Sigma). Sections were incubated at 4 °C overnight with biotinylated HA-binding protein (b-HABP) (1:400, Hokudo), anti-NCAN (sheep, 1:400, R&D), anti-TNC (rat, 1:400, R&D), anti-Pax6 (rabbit, 1:400, Invitrogen), anti-Tbr2 (mouse, 1:400, Wako), anti-Nestin (mouse, 1:400, Millipore), anti-NeuN (rabbit, 1:1000, Proteintech), anti-Ctip2 (rat, 1:400, abcam), and Alexa Fluor 488-conjugated anti-GFP (rabbit, 1:1000, Invitrogen). After washing, sections were incubated with the appropriate Alexa Fluor conjugate secondary antibody and streptavidin (Thermo Fisher), followed by cell nuclear staining with DAPI (0.1 µg/mL). Images were captured with the AX R confocal microscope (Nikon) or LSM 710 NLO confocal microscope (Carl Zeiss). In a three-dimensional reconstruction, images were captured using the Z-stack function of the confocal microscope, and Z-projection and orthogonal views were obtained by ImageJ FIJI software.

## Quantitative reverse transcription (RT)-PCR

Total RNA from the developing mouse cerebral cortex and isolated GFP-positive cells were prepared using the RNeasy mini kit (QIAGEN). cDNA was synthesized using SuperScript III reverse transcriptase (Thermo Fisher). Real-time RT-PCR was performed with the PowerUp SYBR Green Master Mix (Thermo Fisher) in the StepOne PCR system (Thermo Fisher). All primers used are listed in the Key resources table. Relative gene expression was calculated using the comparative cycle threshold (Ct) method. Expression levels were normalized to *Gapdh*.

## In situ hybridization

Target sequences detecting *Ncan* and *Tnc* were amplified from mouse brain cDNA and cloned into the pSPT 19 vector (Roche). All primers used for probe amplification are listed in the Key Resources table. Digoxigenin (DIG)-labeled sense and antisense probes were generated with T7 and Sp6 RNA

polymerase using a DIG RNA labeling kit (Roche) according to the manufacturer's instructions. RNA probes were purified with the RNeasy Mini Kit (Qiagen) and quantified using a Nanodrop (Thermo Fisher Scientific). ICR mouse embryos were removed from pregnant females at E16.5 and perfused with 4% paraformaldehyde in PBS. Brains were post-fixed overnight with 4% paraformaldehyde in PBS and embedded in OCT compound (Sakura Finetek). Frozen sections were cut coronally with a Leica Jung CM3000 cryostat (Leica) at a thickness of 20 µm. Brain sections were fixed with 4% paraformaldehyde in PBS, treated with 1 µg/mL proteinase K, permeabilized with 0.3% Triton X-100 in PBS, and hybridized with 0.5 µg/mL DIG-labeled probes at 70°C overnight in a moist chamber. After washing three times with 0.2×saline-sodium citrate at 70°C for 30 min, sections were blocked with 10% blocking reagent (Roche) at room temperature for 1 hr and incubated with alkaline phosphatase-conjugated anti-DIG antibody (Roche) overnight at 4 °C. The hybridization signal was developed with NBT/BCIP solution (Roche).

## Identification of the 130-kDa band by MALDI-TOF/TOF

Ten E18.5 mouse brains were homogenized in 10 mL of Hanks' Balanced Salt Solution (Gibco) containing 1% Triton X-100, 10 mM HEPES (pH 7.6) (Gibco), and protease inhibitor cocktail (Sigma) and incubated on ice for 1 hr. After centrifugation at 15,000×g at 4 °C for 30 min, the supernatant was applied to a DEAE-Toyopearl 650 M column (1 mL, Tosoh) equilibrated with buffer A (0.1% Triton X-100, 10 mM Tris–HCl, pH 7.5). The column was washed with buffer A containing 0.25 M NaCl and then eluted with buffer A containing 0.5 M NaCl. The eluent was diluted with a four-fold volume of buffer A and digested with 50 milliunits of chondroitinase ABC (Sikagaku Corp) at 37 °C for 4 h. The digest was again applied to a DEAE-Toyopearl 650 M column (0.4 mL) and eluted with buffer A containing 0.25 M NaCl. The eluent was applied to a heparin-agarose column (0.3 mL, Sigma), and the flow-through fraction was collected. After ultrafiltration with the YM-10 membrane (Amicon), the concentrate was separated using 6% SDS-PAGE. After staining with Coomassie Brilliant Blue, the 130 kDa band was cut out as a gel piece and digested with trypsin (Promega) at 37 °C overnight. The recovered tryptic peptides were subjected to nano-LC (DiNa Nano LC system, KYA Technologies), and eluted fractions were spotted on the MALDI plate (AB SCIEX) with α-cyano-4-hydroxyciccanic acid (Shimadzu) as a matrix using the Direct nano LC/MALDI Spotter (KYA technologies). Peptides were analyzed for identification using a MALDI mass spectrometer (TOF/TOF 5800, AB SCIEX). MS/MS data were analyzed using MASCOT software (Matrix Science) as previously described (*Sakurai et al., 2018*).

## Precipitation of HA-binding molecules from mouse brain lysate (*Sugitani et al., 2021*)

One hundred microliters of the E16.5 mouse brain lysate were digested with 1 unit of hyaluronidase from *Streptomyces hyalurolyticus* (Sigma) at 37°C for 1 hr. Digested and undigested samples were incubated with 1.25 µg of b-HABP (Hokudo) or 0.25 ng of biotinylated HA (b-HA) (PG research) at 4 °C overnight, followed by precipitation by mixing with 25 µL streptavidin-coupled magnetic beads (Thermo Fisher) at room temperature for 1 hr. After washing the precipitates with PBST, the supernatant, and precipitates were digested with 5 milliunits of chondroitinase ABC (Sikagaku Corp) at 37°C for 2 hr, denatured by SDS at 95°C for 5 min, and then analyzed by immunoblotting, as described above.

## Construction of GFP-fused NCAN-expressing plasmids

We amplified the full-length *Ncan* sequence by PCR from the mouse brain cDNA library and assembled it into the pCAG-GFP vector using the NEBuilder HiFi DNA Assembly Master Mix (New England Biolabs) or In-Fusion HD Cloning Kit (TaKaRa). The GFP-fused full-length NCAN-expressing plasmid was constructed by inserting the fragment encoding GFP amplified from the pCAG-GFP vector into the central region of NCAN (corresponding to Val641 to Leu644). The plasmids expressing the N- and C-terminal halves of NCAN were constructed by deleting the C-terminal and N-terminal regions from full-length NCAN using the KOD-Plus Mutagenesis Kit (Toyobo).

## Immobilization of GFP-fused NCAN on agarose resin

HEK293 or HEK293T cells were transfected with plasmids expressing GFP-fused NCAN using Lipofectamine 3000 (Thermo Fisher) or PEI Max (Polysciences). After 3 or 4 days, recombinant proteins

were isolated from the culture medium by incubating anti-GFP nanobody-conjugated agarose (Chromotek). In some experiments, we quantified the amounts of GFP-fused full-length, N-terminal half, and C-terminal half of NCAN in the culture medium by immunoblotting with anti-GFP antibody and used equal amounts of GFP-fused NCAN for immobilization.

### Binding of HA and GFP-fused NCAN

GFP fused-NCAN immobilized to agarose was incubated with 2 μg /ml of HA (600–1200 kDa, PG research) in PBS at 4 °C overnight. After washing with PBS, HA bound to agarose resin was released by denaturation at 95 °C for 10 min. HA concentrations were measured by enzyme-linked immunosorbent assays (ELISA), as previously reported (*Egorova et al., 2023*). Briefly, each well of a 96-well plate (Iwaki) was coated with 0.1% HABP (Cosmo-bio) at 4 °C overnight. After washing with PBST, each well was blocked with 1% BLOCK ACE (KAC) in PBS at 37 °C for 2 hr, followed by incubation with 1 mg/mL HA (Wako) in the blocking solution at 37 °C for 1 h. After washing, a mixture of 0.05% b-HABP (Hokudo) and the cerebrum lysates was added to each well and incubated at 37 °C for 1 h. b-HABP was detected by HRP-conjugated streptavidin (1:25000, Wako) and ELISA POD Substrate TMB (Nacalai Tesque).

### Identification of NCAN interactors by LC-MS/MS analysis

GFP-fused NCAN was immobilized on anti-GFP nanobody-conjugated agarose resin. Resin without NCAN was used as a negative control. NCAN-conjugated and unconjugated resins were incubated with embryonic mouse brain lysates at 4 °C overnight. After washing with PBST, precipitated proteins were denatured with 8 M urea, subjected to reductive alkylation by dithiothreitol and iodoacetamide, and then digestion by trypsin (Promega). The resultant peptide fragments were analyzed using the LC-MS/MS system (QTRAP5500, AB Sciex) as previously described (*Sakurai et al., 2018*). MS/MS data were compiled using Mascot Distiller (Matrix Science) and analyzed with MASCOT software (Matrix Science). Data filtering limits were set at a peptide probability ≥95% and a minimum peptide number of 3 per protein.

### Structure prediction using AlphaFold2

Based on the previous rACAN-TNR complex structure (PDB ID 1TDQ; *Lundell et al., 2004*), we selected the amino acid sequences of the C-terminal part of mNCAN (T961-I1224) and FNIII-1–5 of mTNC (S624-E1073). We generated five complex models using AlphaFold2 multimer (v2) (*Evans et al., 2022*) implemented in ColabFold (*Mirdita et al., 2022*). ColabFold ran with default parameters. All structures are visualized using PyMOL v.2.5.0 (Schrödinger, LLC).

### Sequence alignment

Alignments were generated using MAFFT (*Katoh and Standley, 2013*) and visualized using ESPript3 (*Robert and Gouet, 2014*). Amino acid residues involved in the interface region of the predicted mNCAN-TNC complex were analyzed using PDBsum (*Laskowski et al., 2018*) or Protein Interactions Calculator (PIC) webserver (*Tina et al., 2007*).

### Intraventricular injection of hyaluronidase

Pregnant WT mice were anesthetized with isoflurane (3% for induction and 2% for maintenance) during the surgery using an animal anesthetizer (Muromachi). Using a pulled glass capillary, we injected 1–2 μL of hyaluronidase from *S. hyalurolyticus* (1 unit/μl, Sigma) or PBS into the lateral ventricle. Fast Green dye (Wako) was co-injected to visualize successful injections with hyaluronidase. After injections, the uterine horns were placed back into the abdominal cavity to allow further embryonic growth. Coronal sections (thickness of 80 μm) were prepared 2 days after injections and analyzed histochemically to detect HA, NCAN, and TNC as described above. Regarding staining intensity profiles, a 50-μm-wide area spanning the CP to VZ was divided into 100 equal bins, and the fluorescence intensity of each region was assessed using ImageJ FIJI.

### Neuronal migration analysis by Flash Tag (FT) (*Govindan et al., 2018*; *Yoshinaga et al., 2021*)

We injected carboxyfluorescein diacetate succinimidyl ester (1 mM in PBS, Dojindo) into the lateral ventricle at E14.5. After 2 or 3 days, coronal sections (thickness of 80 μm) were prepared, and fluorescent images were captured as described above. The top 10% of cells with high fluorescence intensity were extracted and analyzed using ImageJ FIJI. The cerebral cortex was divided into five equal areas (bins 1–5) from the pia to the ventricle, and the proportion of FT-labeled cells in each bin was calculated. For the experiments investigating the effects of hyaluronidase shown in *Figure 8*, we used WT littermate mice and simultaneously injected 1 unit of hyaluronidase or PBS with the fluorescent dye. In the experiments examining neuronal migration in WT, DKO, NCAN KO, and TNC KO mice shown in *Figure 6*, *Figure 6—figure supplement 2*, and *Figure 7*, we analyzed embryos from different dams with four different genotypes.

### Cortical laminar analysis

Coronal sections (thickness of 80 μm) from WT and DKO mice at 2 weeks of age were immunohistochemically analyzed to identify NeuN-positive and Ctip2-positive neurons. The top 15% of NeuN-positive neurons and the top 4% of Ctip2-positive neurons, exhibiting high fluorescence intensity, were selectively extracted and subjected to analysis using ImageJ FIJI. The cerebral cortex was divided into five equal areas (bins 1–5), and the proportion of NeuN-positive and Ctip2-positive neurons in each bin was calculated.

### In utero electroporation

Mouse embryos at E14.5 or E15.5 were electroporated in utero to label neural progenitor cells in the VZ, as previously described (*Tabata and Nakajima, 2003*; *Takechi et al., 2020*). During surgery, pregnant mice were initially anesthetized with isoflurane (3% for induction and 2% for maintenance) using an animal anesthetizer (Muromachi). Using a pulled glass capillary, we injected 1–2 μL of plasmid solution (1 μg/μL) into the lateral ventricle. The head was clasped with a paired electrode (CUY650P5, NEPA Gene), and square electric pulses (33 V for 50ms, five times in 950 ms intervals) were passed using an electroporator (NEPA21, NEPA Gene). After electroporation, the uterine horns were placed back into the abdominal cavity to allow further embryonic growth.

### Forced expression of GFP-fused NCAN in NCAN KO mice

NCAN KO embryos at E14.5 were electroporated in utero with pCAG-Turbo-RFP alone or with GFP-fused full-length NCAN. Coronal sections (thickness of 80 μm) were prepared 2 days after electroporation and analyzed histochemically to detect TNC and GFP-fused NCAN as described above. Images were captured in the IZ region containing Turbo-RFP-positive neurons using a ×100 magnification objective lens with 3.0 X optical zoom on an AX R confocal microscope (Nikon). A total of 10 optical sections were acquired with a step size of 190 nm. Z-projection views were generated, and the staining intensity of TNC around Turbo-RFP-positive neurons was measured in a 59×59 μm area using ImageJ FIJI.

### Morphological analysis of cortical neurons

Neural progenitor cells in the VZ were transfected with pCAG-mCherry by in utero electroporation at E14. After 53 hr, coronal sections (thickness of 80 μm) were prepared, and fluorescent images were captured as described above. Using ImageJ FIJI, neuronal morphology was evaluated by measuring the length-to-width ratio of labeled neurons and the angle between major neurites and the ventricle surface. Bipolar neurons were defined as cells with a length-to-width ratio greater than three and an orientation angle greater than 70°. In *Figure 9*, we analyzed embryos from different WT and DKO dams to compare neuronal morphology.

### Isolation of migrating neurons from the cerebral cortex

GFP labeling followed by cell sorting was performed as previously reported (*Takechi et al., 2020*). Briefly, neural progenitor cells in the VZ of the cerebral cortex at E14.5 were transfected with pCAG-GFP using in utero electroporation as described above. Embryonic brains were harvested 0.6–4 days later, and GFP-positive regions were dissected from the cerebral cortices under a fluorescence

stereomicroscope (Leica). Cortical fragments were incubated in 0.25% trypsin and mechanically dispersed by gentle pipetting to dissociate single cells. GFP-positive cells were isolated from the single-cell suspension by sorting using a JSAN fluorescence-activated cell sorter (Bay Bioscience).

### Primary cultured cortical neurons

Cover glasses were pre-coated with 20 µg/mL poly-L-ornithine (PLO) (Wako) at 37 °C overnight and then overlaid with 0.1–10 µg/mL of recombinant human TNC (R&D), recombinant human NCAN (R&D), and HA (600–1200 kDa, PG research) at 37 °C for 3 hr. WT and NCAN KO brains were harvested 24 hr after in utero electroporation with pCAG-GFP at E14.5 (*Mubuchi et al., 2022*). GFP-positive regions were dissected from the cerebral cortices under a fluorescence stereomicroscope (Leica). Cortical fragments were incubated in 20 units/mL papain (Worthington Biochemical Corporation) and mechanically dispersed by gentle pipetting. Dissociated cells were plated on ECM-coated cover glasses at a density of 50,000 or 100,000 cells per well (1.9 mm$^2$). Cells were maintained for 1–3 days in Neurobasal Plus Medium (Thermo Fisher) supplemented with 2% B-27 Plus (Thermo Fisher), 5% horse serum (Sigma), and antibiotics (Thermo Fisher). Cultures were fixed with 4% paraformaldehyde, stained with an Alexa 488-conjugated anti-GFP antibody (Thermo Fisher) and anti-Tuj-1 (mouse, 1:1000, Sigma), and imaged as described above. For GFP-labeled neurons, processes longer than 10 µm were measured as neurites and analyzed using NeuronJ (*Meijering et al., 2004*). For neurons stained with anti-Tuj-1 antibody, neurites were analyzed using AutoNeuriteJ (*Boulan et al., 2020*; *Mubuchi et al., 2022*). Neurons without neurites were classified as stage 1, neurons with neurites shorter than 50 µm as stage 2, and neurons with neurites longer than 50 µm as stage 3. Regarding immunoblotting in *Figure 1h* and immunostaining in *Figure 2d*, dissociated cells were plated on PLO-coated cover glasses at 200,000 and 50,000 cells per well, respectively.

### Quantification and statistical analysis

All data are presented as the mean ± SD. Statistical analyses were performed using Python within the Jupyter Notebooks environment and Easy R (*Kanda, 2013*). An unpaired, two-tailed Student's t-test was used to compare results from two groups. Dunnett's test was used to compare multiple experimental groups against the control group. A p-value <0.05 was considered to be significant. Details of statistical analyses, including the statistical tests used, N values, and p values, may be found in the relevant figures and figure legends.

### Materials availability

Plasmids are available upon request to the corresponding author. DKO mice strain is available upon request to the corresponding author by signing a material transfer agreement.

## Acknowledgements

This research was funded by the Ministry of Education, Culture, Sports, Science & Technology, Japan [grant number 15K21067, 18K06130, and 21H05681 to SM] and the LOTTE Foundation [grant name Lotte Shigemitsu Prize to SM]. The authors thank Takeshi Kawauchi for valuable discussion, Makoto Matsuyama for technical advice on i-Gonad, Hitoshi Mori for technical assistance in mass spectrometry, Hidekazu Takahashi for the launch of a computational system for protein structure prediction, and members of Smart-Core-Facility Promotion Organization of Tokyo University of Agriculture and Technology for technical assistance. pCAG-GFP and pCAG-mGFP (Addgene plasmid # 11150 and #14757) were a gift from Connie Cepko. We thank the RIKEN Bioresource Center for *Tnc*-deficient mice (RBRC00169).

## Additional information

### Funding

| Funder | Grant reference number | Author |
| --- | --- | --- |
| Ministry of Education, Culture, Sports, Science and Technology | 15K21067 | Shinji Miyata |
| Ministry of Education, Culture, Sports, Science and Technology | 18K06130 | Shinji Miyata |
| Ministry of Education, Culture, Sports, Science and Technology | 21H05681 | Shinji Miyata |
| Lotte Foundation | Lotte Shigemitsu Prize | Shinji Miyata |

The funders had no role in study design, data collection and interpretation, or the decision to submit the work for publication.

### Author contributions

Ayumu Mubuchi, Data curation, Investigation, Writing - original draft; Mina Takechi, Data curation, Investigation; Shunsuke Nishio, Tsukasa Matsuda, Data curation, Formal analysis, Writing - original draft; Yoshifumi Itoh, Resources, Writing - original draft; Chihiro Sato, Ken Kitajima, Supervision, Validation; Hiroshi Kitagawa, Conceptualization, Supervision, Validation; Shinji Miyata, Conceptualization, Supervision, Funding acquisition, Validation, Writing - original draft, Writing - review and editing

### Author ORCIDs

Shunsuke Nishio  http://orcid.org/0000-0003-3420-2578
Yoshifumi Itoh  http://orcid.org/0000-0002-2128-2823
Hiroshi Kitagawa  http://orcid.org/0000-0002-9307-7079
Shinji Miyata  http://orcid.org/0000-0001-6635-1645

### Ethics

All animal experimental studies were conducted with the approval of the Committee on Animal Experiments of the Graduate School of Bioagricultural Sciences, Nagoya University, and Tokyo University of Agriculture and Technology (Approval Code: R02-111, R03-103, R04-220).

Reviewer #1 (Public Review): https://doi.org/10.7554/eLife.92342.3.sa1
Reviewer #2 (Public Review): https://doi.org/10.7554/eLife.92342.3.sa2
Author Response https://doi.org/10.7554/eLife.92342.3.sa3

## Additional files

### Supplementary files

• MDAR checklist

### Data availability

All data generated or analysed during this study are included in the manuscript and supporting files.

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
