## [Editor Report · eLife assessment]

The **solid** study addresses the role of extracellular matrix (ECM) in neuronal migration. The authors showed that the interaction between the ternary complex formed by tenascin-C, the chondroitin sulfate proteoglycan neurocan, and hyaluronic acid is **important** for the multipolar to bipolar transition in the intermediate zone (IZ) of the developing cortex

---

## [Referee Report · Reviewer #1 (Public Review)]

Summary:

In the present study, authors found the ternary complex formed by NCAN, TNC, and HA as an important factor facilitating the multipolar to bipolar transition in the intermediate zone (IZ) of the developing cortex. NCAM binds HA via the N-terminal Link modules, meanwhile, TNC cross-links NCAN through the CDL domain at the C-terminal. The expression and right localization of these three factors facilitate the multipolar-bipolar transition necessary for immature neurons to migrate radially. TNC and NCAM are also involved in neuronal morphology. The authors used a wide range of techniques to study the interaction between these three molecules in the developing cortex. In addition, single and double KO mice for NCAN and TNC were analyzed to decipher the role of these molecules in neuronal migration and morphology.

Strengths:

The study of the formation of the cerebral cortex is crucial to understanding the pathophysiology of many neurodevelopmental disorders associated with malformation of the cerebral cortex. In this study, the authors showed, for the first time, that the ternary complex formed by NCAN, TNC, and HA promotes neuronal migration. The results regarding the interaction between the three factors forming the ternary complex are convincing.

---

## [Referee Report · Reviewer #2 (Public Review)]

Summary:

ECM components are prominent constituents of the pericellular environment of CNS cells and form complex and dynamic interactomes in the pericellular spaces. Based on bioinformatic analysis, more than 300 genes have been attributed to the so-called matrisome, many of which are detectable in the CNS. Yet, not much is known about their functions while increasing evidence suggests important contributions to developmental processes, neural plasticity, and inhibition of regeneration in the CNS. In this respect, the present work offers new insights and adds interesting aspects to the facets of ECM contributions to neural development. This is even more relevant in view of the fact that neurocan has recently been identified as a potential risk gene for neuropsychiatric diseases. Because ECM components occur in the interstitial space and are linked in interactomes their study is very difficult. A strength of the manuscript is that the authors used several approaches to shed light on ECM function, including proteome studies, the generation of knockout mouse lines, and the analysis of in vivo labeled neural progenitors. This multi-perspective approach permitted to reveal hitherto unknown properties of the ECM and highlighted its importance for the overall organization of the CNS.

Strengths:

Systematic analysis of the ternary complex between neurone, TNC, and hyaluronic acid; establishment of KO mouse lines to study the function of the complex, use of in utero electroporation to investigate the impact on neuronal migration.

---

## [Author Response]

The following is the authors’ response to the original reviews.

**Reviewer 1**
Summary:In the present study, authors found the ternary complex formed by NCAN, TNC, and HA as an important factor facilitating the multipolar to bipolar transition in the intermediate zone (IZ) of the developing cortex. NCAM binds HA via the N-terminal Link modules, meanwhile, TNC cross-links NCAN through the CDL domain at the C-terminal. The expression and right localization of these three factors facilitate the multipolar-bipolar transition necessary for immature neurons to migrate radially. TNC and NCAM are also involved in neuronal morphology. The authors used a wide range of techniques to study the interaction between these three molecules in the developing cortex. In addition, single and double KO mice for NCAN and TNC were analyzed to decipher the role of these molecules in neuronal migration and morphology.Strengths:The study of the formation of the cerebral cortex is crucial to understanding the pathophysiology of many neurodevelopmental disorders associated with malformation of the cerebral cortex. In this study, the authors showed, for the first time, that the ternary complex formed by NCAN, TNC, and HA promotes neuronal migration. The results regarding the interaction between the three factors forming the ternary complex are convincing.

We appreciate the reviewers' positive assessment of our research.

Weaknesses:However, regarding the in vivo experiments, the authors should consider some points for the interpretation of the results:The authors did not use the proper controls in their experiments. For embryonic analysis, such as cortical migration, neuronal morphology, and protein distribution (Fig. 6, 7, and 9), mutant mice should be compared with control littermates, since differences in the results could be due to differences in embryonic stages. For example, in Fig. 6 the dKO is more developed than the WT embryo.

It was challenging to compare double knockout mice with control littermates. When crossing Ncan and Tcn double heterozygous mice, the probability of obtaining double knockout mice is 1/16. Given an average litter size of around 8, acquiring a substantial number of double knockout mice would necessitate an impractical number of breeding pairs. Consequently, we were constrained to use non-littermate control mice. To address potential differences in developmental stages, we analyzed 19-20 embryos obtained from five individuals in each group, demonstrating that the observed differences between the two groups are more substantial than the inherent variability within each group.

The authors claim that NCAM and TNC are involved in neuronal migration from experiments using single KO embryos. This is a strong statement considering the mild results, with no significant difference in the case of TNC KO embryos, and once again, using embryos from different litters.

We agree with the reviewer's comment that a single deletion of TNC has a minimal impact on neuronal migration. We have revised the Results section to reflect the mild nature of the TNC KO phenotype more accurately.

Page 8, line 225: "In NCAN KO mice, a significantly lower percentage of labeled cells resided in the upper layer (Bin2), and more cells remained in the lower layer (Bin5) than in WT mice (Figure 7a). In contrast, the impact of a single deletion of TNC on neuronal cell migration was minimal. Although TNC KO mice exhibited a tendency to have a higher proportion of labeled cells in the lower layer (Bin4) than in WT mice, this did not reach statistical significance (Figure 7a). The delay in neuronal migration observed in the single KO mice was milder when compared to that observed in DKO mice (Figure 6a-c), suggesting that simultaneous deletion of both NCAN and TNC is necessary for a more pronounced impairment in neuronal cell migration."

The measurement of immunofluorescence intensity is not the right method to compare the relative amount of protein between control and mutant embryos unless there is a right normalization.

We agree that measuring immunofluorescence intensity alone is insufficient for comparing the relative amount of protein. In Figure 8, we have employed Western blotting to compare the protein levels, revealing an approximately 50% reduction in NCAN and TNC following hyaluronidase digestion. In Figures 7b and 7c, we demonstrated alterations in the localization patterns of TNC and NCAN in Ncan KO and Tnc KO mice; however, we did not mention their quantity.

Page 7, line 206. "No significant abnormalities were observed in the laminar structure in 4-week-old DKO mice". The authors should be more careful with this statement since they did not check the lamination of the adult cortex. I would recommend staining, control and mutant mice, with markers of different cortical populations, such as Cux1, Ctip2, Tbr1, to asses this point.

In response to the suggestion, we have conducted additional experiments to provide a more detailed examination of the laminar structure in the cerebral cortex. The results have been incorporated into the revised manuscript as follows:

Page 7, line 209: "To investigate the laminar organization of the postnatal cerebral cortex, we analyzed the distribution of NeuN-positive postmitotic neurons in DKO mice at 2 weeks of age. No notable abnormalities were observed in the laminar structure of DKO mice (Figure 6-figure supplement 3a, b). Additionally, the laminar distribution of Ctip2-positive deep layer neurons showed no significant differences between WT and DKO mice (Figure 6-figure supplement 3a, c)."

The authors do not explain how they measured the intensity of TNC around the transfected Turbo-RFP-positive neurons.

We added the following description to the Materials and Methods:

Page 18, line 608: "Images were captured in the IZ region containing Turbo-RFP-positive neurons using a 100X magnification objective lens with 3.0X optical zoom on an AX R confocal microscope (Nikon). A total of 10 optical sections were acquired with a step size of 190 nm. Z-projection views were generated, and the staining intensity of TNC around Turbo-RFP-positive neurons was measured in a 59 × 59 µm area using ImageJ FIJI."

The loading control of the western blots should be always included.

In Figure 6-figure supplement 1, we have incorporated western blot data using a GAPDH antibody as a loading control. We have added an explanation in the figure legend of Figure 3c, stating that we analyzed the same samples as those used in Figure 1e.

For Fig. 3e, I think values are represented relative to E18 instead to P2.

Thank you for pointing that out. As suggested, we have corrected the representation in Fig. 3e to be relative to E18 instead of P2.

I would recommend authors use the standard nomenclature for the embryonic stages. The detection of the vaginal plug is considered as E0.5 and therefore, half a day should be added to embryonic stages (E14.5...).

We have revised our manuscript to designate the detection of the vaginal plug as E0.5, and subsequently, we have adjusted all embryonic stages by adding half a day, such as E14.5.

Fig 10K: I do not see the differences in the number of neurites in the graph.

We have modified the presentation from a box-and-whisker plot to a bar graph to enhance the visibility of differences in the average number of neurites.

Line 37: Not all of the cerebral cortex is structured in 6 layers but the neocortex.

We have changed 'cerebral cortex' to 'cerebral neocortex.'

**Reviewer 2**
Summary:ECM components are prominent constituents of the pericellular environment of CNS cells and form complex and dynamic interactomes in the pericellular spaces. Based on bioinformatic analysis, more than 300 genes have been attributed to the so-called matrisome, many of which are detectable in the CNS. Yet, not much is known about their functions while increasing evidence suggests important contributions to developmental processes, neural plasticity, and inhibition of regeneration in the CNS. In this respect, the present work offers new insights and adds interesting aspects to the facets of ECM contributions to neural development. This is even more relevant in view of the fact that neurocan has recently been identified as a potential risk gene for neuropsychiatric diseases. Because ECM components occur in the interstitial space and are linked in interactomes their study is very difficult. A strength of the manuscript is that the authors used several approaches to shed light on ECM function, including proteome studies, the generation of knockout mouse lines, and the analysis of in vivo labeled neural progenitors. This multi-perspective approach permitted to reveal hitherto unknown properties of the ECM and highlighted its importance for the overall organization of the CNS.Strengths:Systematic analysis of the ternary complex between neurons, TNC, and hyaluronic acid; establishment of KO mouse lines to study the function of the complex, use of in utero electroporation to investigate the impact on neuronal migration;

We appreciate the reviewers' insightful comments.

Weaknesses:The analysis is focused on neuronal progenitors, however, the potential impact of the molecules of interest, in particular, their removal on differentiation and /or survival of neural stem/progenitor cells is not addressed. The potential receptors involved are not considered. It also seems that rather the passage to the outer areas of the forming cortex is compromised, which is not the same as the migration process. The movement of the cells is not included in the analysis.

In this study, we demonstrated that the ternary complex of NCAN, TNC, and HA is predominantly localized in the subplate/intermediate zone. This region lacks neural stem/progenitor cells but serves as the initiation site for the radial migration of postmitotic neurons. Consequently, our study focused on the role of the ternary complex in neuronal migration and polarity formation.We acknowledge that we did not investigate in-depth the potential effects of ECM perturbation on the differentiation and survival of neural stem/progenitor cells. However, as highlighted by the reviewer, it is important to explore the effects on neural stem/progenitor cells. To address this concern, we analyzed Pax6-positive radial glial cells and Tbr2-positive intermediate progenitor cells in the ventricular zone of wild-type and Ncan/Tnc double knockout (DKO) mice. Immunohistochemical analysis revealed no significant differences between WT and DKO mice (Figure 6-figure supplement 4a). Furthermore, the morphology of nestin-positive radial fibers exhibited no distinguishable variations between WT and DKO mice (Figure 6-figure supplement 4b, c).

(1) In the description of the culture of cortical neurons the authors mentioned the use of 5% horse serum as a medium constituent. HS is a potent stimulus for astrocyte differentiation and astrocytes in vitro release neurocan. Therefore, the detection of neurocan in the supernatant of the cultures as shown in Figure 1h might as well reflect release by cultivated astrocytes.

As pointed out by the reviewer, Figure 1h did not conclusively demonstrate that neurons are the sole source of NCAN production. Indeed, in situ hybridization analysis revealed the widespread distribution of Ncan mRNA throughout the cerebral cortex (Figure 2a). This result suggests that the production of NCAN involves not only neurons but also other cell populations, including radial glial cells and astrocytes. While we acknowledge the potential contribution of other cell types to NCAN production, Ncan expression by neurons during radial migration is a crucial aspect of our findings (Figure 1i, j). We have revised the manuscript as follows:

Page 5, line 111: "This result suggested the secretion of NCAN by developing neurons; however, we cannot rule out the involvement of coexisting glial cells in the culture system. To investigate the expression of Ncan mRNA during radial migration in vivo, we labeled radial glial cells in the VZ with GFP through in utero electroporation at E14.5 (Figure 1i, Figure 1-figure supplement 1)."

(2) It is known that neurocan in vivo is expressed by neurons, but may be upregulated in astrocytes after lesion, or in vitro, where the cells become reactive.

We have incorporated the following description into the discussion:

Page 11, line 359: "Previous studies have reported an upregulation of NCAN and TNC in reactive astrocytes, indicating the potential formation of the ternary complex of NCAN, TNC, and HA in the adult brain in response to injury (Deller et al., 1997; Haas et al., 1999)."

(3) Do NCAN KO neurons show an increase in neurite growth on the TNC substrates? The response on POL was changed (Fig. 10h-k), but the ECM substrates were not tested with the KO neurons.

The impact of ECM substrates on NCAN KO neurons has not been investigated, and this remains an avenue for further exploration in our ongoing research. Future studies aim to elucidate the NCAN-TNC connection by identifying TNC cell surface receptors and unraveling the subsequent intracellular signaling pathways.

(4) Do the authors have an explanation for why the ternary complex is concentrated in the SP/IZ zone?

In the mature brain, hyaluronan acts as a scaffold that facilitates the accumulation of ECM components, including proteoglycans and tenascins around neurons. Therefore, it is conceivable that the ECM components bind to hyaluronan in the embryonic brain, resulting in its accumulation in the subplate/intermediate zone. In support of this hypothesis, enzymatic digestion of hyaluronan in the subplate/intermediate zone led to the disappearance of TNC and NCAN accumulation (Figure 8a-c). This result may account for the disparity observed, where Tnc mRNA is expressed in the ventricular zone while the TNC protein localizes to the subplate/intermediate zone.

(5) Are hyaluronic acid synthesizing complexes (HAS) concentrated in the SP/IZ?

According to the reviewer's comment, we have investigated the localization of Has2 and Has3 mRNA using in situ hybridization. However, due to the relatively low expression levels of these enzymes, we encountered challenges in obtaining clear signals (Author response image 1). Further research is needed to understand the mechanisms behind the localization of hyaluronan in the intermediate zone.

**Author response image 1. sa3fig1:** In situ hybridization analysis of Has2 and 3 mRNA on the E16. 5 cerebral cortex. Upper images show results of in situ hybridization using antisense against Has2 and 3. Lower images are in situ hybridization using sense probes as negative controls.

(6) CSPGs as well as TNC are part of the neural stem/progenitors cell niche environment. Does the removal of either of the ECM compounds affect the proliferation, differentiation, and/or survival of NSPCs, or their progeny?(7) This question relates to the fact that the migration process itself is not visualized in the present study, rather its outcome - the quantitative distribution of labeled neurons in the different bins of the analysis. This could also derive from modified cell numbers.

As pointed out by the reviewer, previous studies have shown the role of CSPGs and TNC as components of the neural stem/progenitor cell niche (see reviews by Faissner et al., 2017; Faissner and Reinhard, 2015). However, as mentioned in Response #2, based on our analyses, we did not observe a reduction in neural stem/progenitor cells in NCAN/TNC double-knockout mice. While we cannot precisely explain this discrepancy, it is worth noting that many past studies evaluated the activities of the ECM molecules in in vitro systems such as neurospheres. The observed differences may stem from variations in experimental systems.

(8) What is the role of the ECM in the SP/IZ area? Do the cells need the ECM to advance, the reduction would then leave the neuronal progenitors in the VZ area? This somehow contrasts with interpretations that the ECM acts as an obstacle for neurite growth or cell migration, or as a kind of barrier.

The role of the ECM is multifaceted, with certain ECM molecules known to inhibit neurite outgrowth while others facilitate it. Additionally, the effects of ECM can vary depending on the cell type. It is established that after migrating neurons adhere to radial fibers, they utilize these fibers as a scaffold to migrate toward the cortical surface. However, in the subplate/intermediate zone, migrating neurons have not yet adhered to radial fibers. This study provides evidence that multipolar neurons undergo morphological changes into bipolar cells with the assistance of the NCAN, TNC, and HA complex. Subsequently, this facilitates their movement along radial fibers.

(9) A direct visualization of the movement of neural progenitors in the tissue as has been for example performed by the Kriegstein laboratory might help resolve some of these issues.

As suggested by the reviewer, utilizing live imaging techniques to directly observe the movement of neural progenitors within the tissue is indeed a powerful tool. We recognize the significance of addressing these points in future research.